# LARG$^2$, Language-based Automatic Reward and Goal Generation

## Abstract

Robotic tasks currently addressed with reinforcement learning such as locomotion, navigation, and manipulation are challenged with the problem of defining reward functions to maximize and goals to reach. Alternative methodologies, like imitation learning, often require labor-intensive human annotations to produce datasets of task descriptions associated with trajectories. As a response, this paper introduces "Language-based Automatic Reward and Goal Generation" (LARG$^2$), a framework that harnesses code generation capabilities of LLMs to enables the conversion of text-based task descriptions into corresponding reward and goal-generation functions. We leverages Chain-of-thought mechanisms and the common-sense knowledge embedded in Large Language Models (LLMs) for this purpose. It is complemented by automatic error discovery and correction mechanisms. We validate the effectiveness of LARG$^2$ by conducting extensive experiments in the context of robotic manipulation demonstrating its ability to train and execute without human annotation of any kind.

## 1 Introduction

The statistical learning approach to robot control has emerged with the potential of revolutionizing various industries, spanning from manufacturing to healthcare. Various preliminary approaches, such as imitation learning (Tai et al., 2016; Kumar et al., 2022), transfer learning (Stüber et al., 2018; Wiese et al., 2021; Weng et al., 2020), and interactive learning (Kelly et al., 2018; Chisari et al., 2021; Faulkner et al., 2020), have been proposed for that matter.

In the field of robotic manipulation, decision models are currently evolving from the traditional optimal control approaches towards policy learning through Multi-task and Goal-Conditioned Reinforcement Learning (Huang et al., 2022b). Following this line of work, multi-modal task definition (Jiang et al., 2022; Shah et al., 2022), associated with reasoning and action planning abilities facilitated by Large Language Models (LLMs) (Huang et al., 2022a), have enabled agents to adapt to real-world uncertainty which is hardly handled with traditional robotic control. However, **the difficulties of connecting textual descriptions of tasks with their associated goals and reward functions have led to unscalable solutions involving labor-intensive annotation practices**.

Motivated by these observations, we introduce LARG$^2$, Language-based Automatic Reward and Goal Generation. For a given sequential decision task described using natural language, our method automates the generation of either goals or associated reward functions depending on the learning scheme. It leverages the common-sense and reasoning capabilities offered by recent LLMs in terms of text understanding and source code generation. In the context of robotic manipulation, our approach samples goals conditioned by a task description to train a corresponding policy using Goal-Conditioned Reinforcement Learning (GCRL). Following this idea, we generate executable reward functions to train corresponding policies using Multi-Task Reinforcement Learning (MTRL), assuming task descriptions are given as input to the policy. Finally, we evaluate these two settings of LARG$^2$ over a set of language-formulated tasks in a tabletop manipulation scenario.

## 2    PRELIMINARIES: REINFORCEMENT LEARNING FOR ROBOTIC MANIPULATION

Reinforcement Learning deals with an agent performing sequences of actions in a given environment to maximize a cumulative sum of rewards. Such problem is commonly framed as Markov Decision Processes (MDPs): $M = \{S, A, T, \rho_0, R\}$ (Sutton & Barto, 2005; Mnih et al., 2016; Lillicrap et al., 2016). The agent and its environment, as well as their interaction dynamics, are defined by the first components $\{S, A, T, \rho_0\}$, where $s \in S$ describes the current state of the agent-environment interaction and $\rho_0$ is the distribution over initial states. The agent interacts with the environment through actions $a \in A$. The transition function $T$ models the distribution of the next state $s_{t+1}$ conditioned with the current state and action $T : p(s_{t+1}|s_t, a_t)$. Then, the objective of the agent is defined by the remaining component of the MDP, $R : S \to \mathbb{R}$. Solving a Markov decision process consists in finding a policy $\pi : S \to A$ that maximizes the cumulative sum of discounted rewards accumulated through experiences.

In the context of robotic manipulation, a task commonly consists in altering the environment into a targeted state through selective contact (Gu et al., 2017). Naturally, tasks are expressed as $g = (c_g, R_G)$ pair where $c_g$ is a goal configuration such as Cartesian coordinates of each element composing the environment or a textual description of it, and $R_G : S \times G \to \mathbb{R}$ is a goal-achievement function that measures progress towards goal achievement and is shared across goals. A goal-conditioned MDP is defined as : $M_g = \{S, A, T, \rho_0, c_g, R_G\}$ with a reward function shared across goals. In multi-task reinforcement learning settings, an agent solves a possibly large set of tasks jointly. It is trained on a set of rewards associated with each task. Finally, goals are defined as constraints on one or several consecutive states that the agent seeks to satisfy (Plappert et al., 2018; Nair et al., 2018; OpenAI et al., 2021).

## 3    RELATED WORK

### 3.1    CHALLENGES OF REWARD DEFINITION AND SHAPING

A sequential decision task which is not solved through imitation but reinforcement requires defining an informative reward function to enable the learning paradigm. Reward shaping consists in manually designing a function incorporating elements from domain knowledge to guide policy search algorithms. Formally, this can be defined as $R' = R + F$, where $F$ is the shaping reward function, and $R'$ is the modified reward function Dorigo & Colombetti (1994); Randløv & Alstrøm (1998). As a main limitation, a reward function needs to be crafted for each task. For instance Brohan et al. (2022) leveraged large number of human demonstrations and specific handcrafted definitions of tasks to train a robotic transformer. However, as MTRL aims at dealing with a large set of goals and tasks to implement, such an approach becomes hardly scalable. In this work, we study how to leverage the common-sense and prior knowledge embedded in LLMs to automate the textual paraphrasing of task description and the generation of associated reward functions.

### 3.2    LARGE LANGUAGE MODELS FOR CONTROL

The use of Large Language Models to control autonomous agents has recently started to be investigated. Shah et al. (2022) has combined a text encoder, a visual encoder, and visual navigation models, to provide text-based instructions to a navigating agent. This idea has been further developed in Huang et al. (2022b) using LLM capabilities to support action planning, reasoning, and internal dialogue among models for manipulation tasks. Similarly, Liang et al. (2022) proposes to use LLMs to transform textual instructions into a code-based policy. Unfortunately, it involves an interactive design process for a hard-coded policy, rather than a task-conditional learning process. In contrast, our method, which also relies on a specific prompt design, allows the agent to learn new skills through goal generation and automatic reward shaping.

Along this line, Colas et al. (2020b) proposes to derive goals from a textual description of the task. However, the language remains limited to the logical descriptions of the expected configuration of the scene and the goal is reduced to a finite set of eligible targets. In contrast, our approach allows using natural language beyond logical forms, grounded with reasoning capabilities and enriched with common-sense captured in large pre-trained language models. Also related, Colas et al. (2020a)

proposes to train a conditional variational auto-encoder to create a language-conditioned goal generator. However, it assumes the existence of pre-trained goal-conditioned policies and no LLM is considered to achieve this objective.

### 3.3 IMPROVING GENERATION WITH CHAIN OF THOUGHT

To address LLMs limitations such as hallucination, lack of consistency or lack of grounding several works attempt to enhance the alignment of generated answers with expected behavior or constraints. The "Chain-of-Thought" (CoT) approach (Wei et al., 2023) aims to influence text generation by using a sequence of intermediate reasoning steps as part of the LLM prompt, thereby promoting a consistent generation path. It involves providing examples of expected reasoning behavior as part of the prompt (Wei et al., 2022; Wang et al., 2023b; Wu et al., 2023; Diao et al., 2023). This approach has recently shown successes, particularly in handling complex queries such as mathematics reasoning questions (Imani et al., 2023).

In a subsequent study (Wang et al., 2023a), the authors conducted an analysis of the influence of example composition on reasoning consistency and accuracy. They underscored the importance of example relevance in provided reasoning steps to achieve accurate answers.

In our approach, we hypothesize that CoT presents a promising mechanism to enhance LLM's code generation capabilities. To this end, we leverage existing code repositories to support reasoning for goal and reward function generation.

### 3.4 CONCURRENT WORK

Recently, a method for the generation of reward functions in the context of robotic skill learning as been introduced in Yu et al. (2023). However, this concurrent work exclusively focuses on generating goal poses to complement existing reward functions in the context of robotic manipulation and quadruped's pose control. The generation of reward functions is not addressed in this work. Furthermore, neither code correction nor Chain-of-though mechanism are considered to guide and possibly fix code generation.

## 4 LARG$^2$, LANGUAGE-BASED AUTOMATIC REWARD AND GOAL GENERATION

Our method, illustrated in figure 1, translates textual task descriptions into both goal and reward functions to enable scalable training of goal conditioned (GCRL) and muti-tasks reinforcement learning (MTRL) policies. It is composed of three sequential steps. The initial one is responsible for gathering input for the second step, which carries out code generation. The final step assesses and, if necessary, correct generated functions through a feedback loop. Once validated, the code is used with standard off-the-shelf GCRL or MTRL frameworks. For the MTRL scenario, the second step also encodes textual task descriptions into an embedding vector which is appended to the state vector to align policies with task definitions.

### 4.1 ELEMENTS OF THE PROMPT

Our first step consists in collecting inputs for building a dedicated prompt ($P_2$) to condition code generation. One main element is the task definition ($T$) so to automate the production of a large training set we leverage paraphrasing capabilities of a pre-trained LLM ($L1$) to generate variations from a single description. We use a prompt ($P_1$) such as: "*Generate* n *paraphrases for the task bellow:*". It produces a set of semantically similar tasks without handcrafting a whole collection of tasks such as, $L1(P, T) \rightarrow \{T_1, ..., T_n\}$.

Supplemental code examples ($X$) can also be collected to complement the main prompt ($P_2$) enabling a Chain-of-Thought (CoT) mechanism to guide the code production process. For this, we assume the availability of code repositories such as Github[1]. These repositories need to possess adequate documentation, and the code should be commented. Naturally, it is preferable for this code

---

[1]https://www.github.com

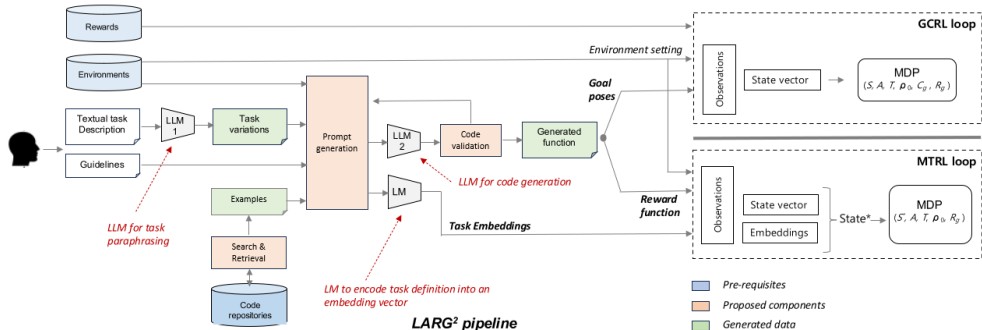

Figure 1: LARG$^2$ transforms a textual task description into either 1) a goal to be used as input of a given reward function for GCRL, or 2) a reward function for MTRL. We use pre-trained and instructed LLMs with dedicated prompts for our generation procedures. For GCRL, the goal is appended to the state description given as input to the policy. For MTRL, the text-based task description is encoded using a pre-trained language model to complement the state vector. Optionally, supplemental code examples can be searched and retrieved to complement the prompt therefore leveraging a Chain-of-Through mechanism. A code validation loop is provided to ensure that generated functions can be properly executed within GCRL or MTRL frameworks.

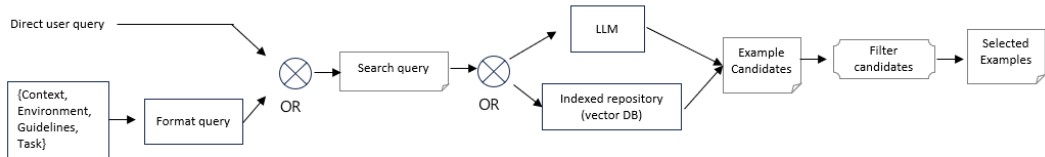

Figure 2: LARG$^2$ leverages context, environment, guidelines and task descriptions to query either the parametric memory of a LLM or a code database to retrieve function examples. It serves as additional context, enriching a dedicated prompt used to convert textual task descriptions into goal poses or reward functions.

to be correct, although Wang et al. (2023a) has shown that even invalid examples used in a CoT mechanism can still yield valid answers.

As illustrated in figure 15, we propose two viable options. Either code repositories are part of a dataset used to train a LLM, in which case the information is embedded within and accessible from the model's parametric memory, or they are independently indexed. The latter option allows to extend the LLM's background knowledge with external information, which may be more relevant for a specific application.

In the following description, we focus on the latter approach although we also test parametric memory during our experiments. First, we segment each code file from a repository into a set of functions and each function is indexed individually. This indexing process ($I$) combines information from multiple sources, including the readme.md file ($R$), the function's signature ($S$), its docstring ($D$), and its code ($C$). This aggregation can be represented as: $R, S, D, C \rightarrow F$, where $F$ represents the indexed function. The result is encoded into a collection of embeddings and stored within a Vector database for semantic retrieval.

## 4.2 GENERATING GOAL AND REWARD FUNCTIONS

The second stage use a dedicated prompt ($P_2$) with ad-hoc parameters to query a LLM ($L_2$) for generating either goal or reward functions. This prompt, illustrated in figure 3, is composed of $\{T, G, E, X\}$ where $T$ and $G$ are provided by the user, or through paraphrasing, $E$ from pre-

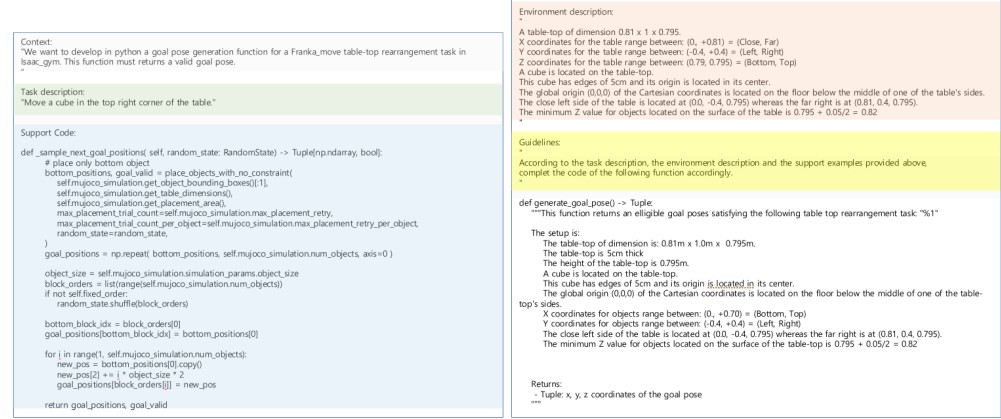

Figure 3: The prompt is composed of a set of parts describing the general context which is highlighted in grey, supporting code examples in blue, the environment description in orange, the task description in green, guidelines in yellow and the description of the function signature to be used as template for code generation.

requisite dataset and, optionally, $X$ either from the parametric memory of a LLM or from queries to an ad-hoc code database.

$C$ is the high level description of the objective such as "*We aim to develop a Python function for generating goals for a Franka-Move tabletop rearrangement task within IsaacGym*". $T$ is the task description and $E$ provides critical information defining the action space. It includes details such as the dimensions, and locations of objects involved in the experiments. $X$ is an optional list of code examples to guide intermediate reasoning steps of the LLM. Guidelines, $G$, reflect a comprehensive summary referencing preceding sections. It consolidates the list of elements or constraints that must be taken into account when generating the code. For CoT, its purpose is to provide the reasoning schema required to generate a more relevant function. $S$ is the signature of the function that needs to be completed, along with its docstring. This specification ensures that the generated function aligns with specific requirements, enabling it to be executed seamlessly within a larger GCRL or MTRL framework. Figure 15 illustrates the search and retrieval process for supplemental examples ($X$).

### 4.2.1 AUTOMATIC GENERATION OF GOALS FOR GCRL APPLICATIONS

In the context of tabletop manipulation scenarios, a task consists in re-arranging a set of objects composing the scene. In such a case, goals are objects' Cartesian coordinates. In a GCRL settings, these goals parameterize a reward function which, for instance, incorporates environment-dependent reward terms and Euclidian distance between the current pose of the objects and the target pose. Therefore, goals generated by LARG$^2$ are used to compute the reward signal at each step. The prompt $p$, described in previous section, allows to generates a function $F$ such as $L2(\{T, G, E, X\}, P_2) \rightarrow F$ to set goal values.

### 4.2.2 AUTOMATIC GENERATION OF REWARD FUNCTIONS FOR MTRL APPLICATIONS

The second utilization of LARG$^2$ generates the implementation of a reward function. While Large Language Models can support the full generation of complex reward functions ($R$), we propose to simplify the generation by identifying different parts in such a function, some being task-independent ($I$) and others closely related to the task definition ($D$) so that $R$ is a composition of both parts, $R = I + D$. In robotic manipulation, common task-independent components address bonuses for lifting the objects or penalties for the number of actions to achieve a given purpose. Task-dependent components, which are driven by the textual task description, align constraints with penalties ($N$) and guidelines with bonuses ($B$). Both components are combined in a global reward function.

To compose this global reward function, we consider the existence of predefined categories of tasks with their environments, formalized using languages such as YAML [2] or Python, providing independent reward components ($I$) available in repositories like Isaac_Gym [3]. In that respect, the search and retrieval step allows to collect reward components as examples to support full reward generation.

For the task dependant part of the reward, we leverage the generation capability of the LLM (L2) to map task descriptions into bonuses ($B$) and penalties ($N$) so that:

$$R = I + \sum_{i=1}^{n} \alpha_i.B_i + \sum_{j=1}^{m} \beta_j.N_j$$

Weights ($\alpha$ and $\beta$) associated with these parameters could be adjusted in an optimization loop.

### 4.2.3 TASK-ENCODING AND POLICY

For GCRL, the input of the policy is composed with the environment state and the goal generated by LARG$^2$. For MTRL, the goal of each task is replaced by a textual description of the task. We use Google T5 (Raffel et al., 2020a;b) as pre-trained text encoder to encode the text into an embedding vector. This vector is added to the state vector, along with proprioception and exteroception data, in the training phase to label tasks. This approach allows to use textual descriptions of tasks as input to neural policies such as what is proposed by (Jiang et al., 2022).

### 4.3 CODE VALIDATION AND AUTO-CORRECTION

Naturally, the generated code can not be guarantee in terms of code validity or outcomes. As a consequence, we automate iterations, emphasizing the elements that need to be modified until the result converges toward expectations. The errors which are commonly encounter correspond to under-specified elements in the original prompt or from LLM limitations such as hallucinations (Ji et al., 2022). So, we finalize the code generation with an automatic validation step described in Figure 4 which exploits the output of the Python interpreter.

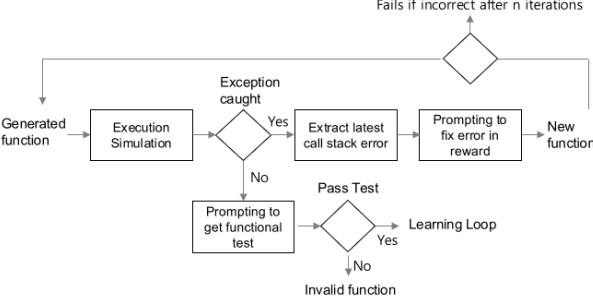

Figure 4: The code correction loop uses the exceptions raised during execution to request modifications. Then, a functional test is generated before moving to the learning loop.

For validation purposes, we execute the generated code using placeholder variables. If the code fail, we catch the exceptions raised by the Python interpreter filtering the thread of exceptions to keep the latest stack and use the error message to fill a prompt requesting code modifications. As illustrated in Figure 5, our prompt contains (1) a header which requests the LLM to fix the raised exception, (2) the text of the raised exception, and (3) the code of the incorrect function. Several iterations can be performed until the code can be properly executed.

Once the generated function satisfies the code correction step, we use another prompt to generate a functional test to evaluate this first function as detailed in section A.1.5 of the appendix. This step filters out potentially incorrect code prior to running the training loop. This prompt, illustrated in Figure 6, is composed of (1) a header requesting the LLM to generate a functional test, (2) a list of guidelines conditioning the test, and (3) the generated function.

---

[2]https://yaml.org/
[3]https://developer.nvidia.com/isaac-gym

```
Could you please fix the error:
'line 38, in compute_franka_reward
RuntimeError: Expected all tensors to be on the same device, but found at least
two devices, cuda:0 and cpu!
'
in the following function implementation:
import torch
from torch import Tensor
from typing import Tuple

def compute_franka_reward(object_pos: Tensor, lfinger_grasp_pos: Tensor, rfinge
r_grasp_pos: Tensor) -> Tuple[Tensor, Tensor]:
                              ...
```

Figure 5: Prompt for the automatic code correction step which contains the error message and the code to be improved in blue.

```
Update the following python script with functional tests for the reward
function 'compute_franka_reward'.
Rewards tests should only validate cases when they should be positive (>=0) or
negative (<=0).
Successes should be tested against 1 or 0 values.
Do no add any explanation text.
Return the same script plus what you have inserted.

import torch
from torch import Tensor
from typing import Tuple

def compute_franka_reward(object_pos: Tensor, lfinger_grasp_pos: Tensor, rfinge
r_grasp_pos: Tensor) -> Tuple[Tensor, Tensor]:
                              ...
```

Figure 6: Prompt requesting the generation of a functional test for the provided function, in blue.

## 5 EXPERIMENTS

Three experiments are designed in the context of robotic manipulation using a table top scenario to answer the following questions: Can we generate valid goal positions from textual task descriptions for GCRL settings; Can we automatically generate reward functions to train multi-task policies; How code examples enhance the relevance of generated functions?

In the GCRL case, we evaluate goal generation on a series of 8 tasks involving a single object, and 4 tasks involving a set of 3 objects. The list of tasks is detailed in Table 2. In the MTRL case, we address the generation of reward functions for 9 manipulation tasks detailed in Table 5 of the appendix. As a source of predefined environments and reward functions for the GCRL experiment, we use Pick and Place scenarios and repositories defined in the IsaacGym repository designed for a Franka Emika Panda robot arm [4].

### 5.1 CHAIN-OF-THOUGHT FOR CODE GENERATION

As a first experiment, we test the compliance of the generated poses against specifications provided in task descriptions. Also, we evaluate the impact of the additional code examples on the relevance of generated goals. For this, we prompt a pretrained LLM to generate a short set of supplemental python functions. A list of these prompts is provided in section A.2.4 of the appendix.

In this evaluation, we use three LLMs: GPT4 (GPT) [5], Hyper Clova X (HCX) [6], and StarCoder (SC)[7]. These models are used in the LARG$^2$ pipeline either in a straightforward manner without supplemental examples in the prompt, or with retrieved functions (RF) provided as examples.

| Task | GPT | GPT+RF | HCX | HCX+RF | SC | SC+RF |
|---|---|---|---|---|---|---|
| Move a cube in the top right corner of the table. | **0.8** | 0.75 | 1 | 0.25 | 0 | 0 |
| Lift the cube 15cm above the table. | 0.8 | **0.9** | 0.8 | 0 | 0 | 0 |
| Take the cube and move it to the left side of the table. | **1** | 0.75 | **1** | 0.5 | 0.25 | 0 |
| Take the cube and move it closer to the robotic arm. | 0.4 | 0.5 | 0 | **1** | 0 | 0 |
| Move the cube 20cm to the left of its initial position. | 0.5 | **0.75** | 0.5 | 0 | 0 | 0 |

Table 1: Performance comparison between three LLMs: GPT4 (GPT), HyperClovaX (HCX), and StarCoder (SC).

Inspired by the results presented in table 1 which highlight a positive impact of supplemental examples using the GPT4 model, we apply this model in subsequent experiments to test LARG$^2$ for Goal-Conditioned and Multi-Task Reinforcement Learning.

### 5.2 LARG$^2$ FOR GOAL-CONDITIONED REINFORCEMENT LEARNING

For this experiment, we use a neural policy trained beforehand using Proximal Policy Optimization (Schulman et al., 2017). The policy takes as input the position and velocity of each joint of the robot and the respective pose of the objects composing the scene. The policy triggers joint displacement

---

[4]https://www.franka.de/

[5]https://openai.com/gpt-4

[6]https://clova.ai/hyperclova

[7]https://huggingface.co/blog/starcoder

in a $\mathbb{R}^7$ action space. The goal information, generated by LARG$^2$ is used as additional input to the policy.

Regarding prompting, we create a dedicated code database to support search and retrieval for supplemental examples using The Stack [8] which is a database that contains 6TB of source code files covering 358 programming languages build as part of the BigCode project [9]. For the sake of performance, we keep only Python files from repositories related to robot learning for manipulation tasks. We use text-based information found in markdown files associated with each repository for this filtering process. Once filtered, we index and store this dataset in a vector database, ChromaDB [10]. Repository descriptions, comments and function names are encoded using SentenceTransformer [11].

To evaluate the benefit of the search and retrieval process, we test the influence of two parameters: the number of provided examples and the alignment, or lack of, between the names of these functions and the name of the targeted one as defined in the signature ($S$) part of the prompt.. Specifically, we include either one, two, or three functions as examples and explore modifications of the supplemental function names to match the name of the expected function. This particular modification is inspired by an observation highlighted in Wang et al. (2023a) which underscores the importance of name coherence within the Chain-of-Thought mechanism.

In Table 2, we evaluate the validity of generated goal poses with respect to textual task descriptions. We compare LARG$^2$without supplemental examples (L) with the retrieval augmented version including 2 and 3 top ranked functions (I_2 and I_3), only the best function (I_b) and a random selection among the top 4 excluding the top one. Similarly we replicate the experiment with modifications of supplemental function names to match the targeted function (M_2, M_3, M_b, M_r).

| Task | L | I_b | I_r | I_2 | I_3 | M_b | M_r | M_2 | M_3 |
|---|---|---|---|---|---|---|---|---|---|
| Move a cube in the top right corner of the table. | 0.75 | 0.7 | **0.9** | **0.9** | 0.4 | 0.5 | **0.9** | **0.9** | 0.25 |
| Lift the cube 15cm above the table. | **1.0** | **1.0** | 0.8 | 0.9 | 0.8 | 0.0 | **1.0** | **1.0** | **1.0** |
| Take the cube and move it to the left side of the table. | **1.0** | **1.0** | **1.0** | **1.0** | **1.0** | **1.0** | 0.3 | 0.6 | **1.0** |
| Take the cube and move it closer to the robotic arm. | 0.5 | 0.6 | 0.6 | 0.4 | 0.3 | 0.3 | **0.7** | 0.3 | **0.7** |
| Lift the cube 20cm above the table and 15 cm ahead. | 0.5 | **1.0** | **1.0** | **1.0** | **1.0** | **1.0** | **1.0** | **1.0** | **1.0** |
| Push a cube 10cm to the right and 10cm backward. | 0.2 | 0.5 | 0.2 | 0.2 | 0.5 | **0.6** | 0.5 | 0.5 | 0.5 |
| Grab a cube and lift it a bit and move it a bit ahead. | **1.0** | 0.5 | 0.7 | 0.8 | 0.7 | 0.7 | 0.7 | 0.8 | 0.9 |
| Move the cube at 20cm to the left of its initial position. | 0.5 | 0.6 | 0.7 | 0.8 | **0.9** | 0.7 | 0.5 | 0.5 | 0.5 |
| Move one cube to the left side of the table, another one to the right side of the table, and put the last cube at the center of the table. | 0.9 | 0.7 | 0.8 | 0.5 | 0.4 | **1.0** | 0.6 | 0.7 | 0.7 |
| Move the three cubes so they are 10 cm close to one another. | 0.9 | **1.0** | **1.0** | 0.3 | 0.2 | **1.0** | **1.0** | 0.2 | 0.2 |
| Move the three cubes on the table so that at the end they form a right-angled triangle. | **1.0** | **1.0** | 0.9 | 0.2 | 0.1 | **1.0** | **1.0** | 0.3 | 0.2 |
| Reposition the three cubes on the table such that they create a square, with the table's center serving as one of the square's corners. | 0.8 | 0.2 | **0.9** | 0.2 | 0.1 | 0.8 | **0.9** | **0.9** | 0.8 |

Table 2: Evaluation of LARG$^2$performance for goal pose generation according to various configurations of the code example part of the prompt.

This experiment demonstrates both the capability of LARG$^2$ to generate goals that match requirements as defined in task descriptions and the positive influence of additional code samples on the accuracy of generated functions. Furthermore, it reveals that the naming of functions has minimal impact on performance.

## 5.3 LARG$^2$ FOR MULTI TASK REINFORCEMENT LEARNING

In this experiment, we train a policy using Proximal Policy Optimization with default Franka Move parameters. The policy takes as input the task description which is encoded using a pre-trained Google T5-small language model Raffel et al.. For each task, we use the *[CLS]* token embedding computed by the encoder layer of the model which is defined in $\mathbb{R}^{512}$. We concatenate this embedding with the state information of our manipulation environment defined in $\mathbb{R}^7$ and feed it into a stack of fully connected layers used as policy. This policy is composed of 3 layers using respectively, $\{512, 128, 64\}$ hidden dimensions. Alternately, as suggested by Jiang et al. (2022), we tested feeding the token embedding into each layer of the stack instead of concatenating it as input but we did not observe improvements.

---

[8]https://huggingface.co/datasets/bigcode/the-stack

[9]https://www.bigcode-project.org/

[10]https://www.trychroma.com/

[11]https://www.sbert.net/

For the reward generation process, we first set the task-independent reward component leveraging rewards available from IsaacGym for pick and place manipulation. This component handles gripper finger distance to the object, bonuses for lifting the object and penalties for the number of actions to reach the objective. This component, which is therefore common to each task, is not generated. It is added to the task dependant reward generated by LARG$^2$ for each task. Details about this process are further discussed in section A.1.3 of the appendix. Reward functions apply goal poses generated according to the task to compute related scores. Figures 7 and 8, respectively present generated goal positions for 9 manipulation tasks, detailed in the appendix, and the success rate of subsequently trained policies.

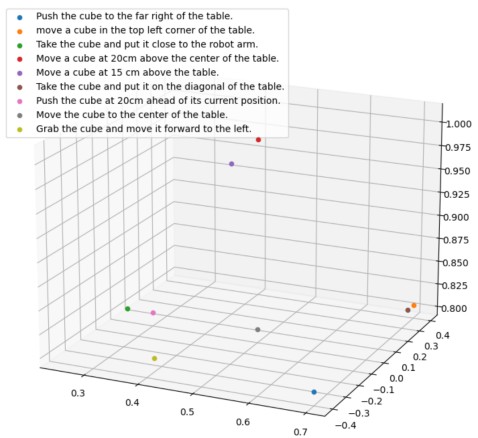

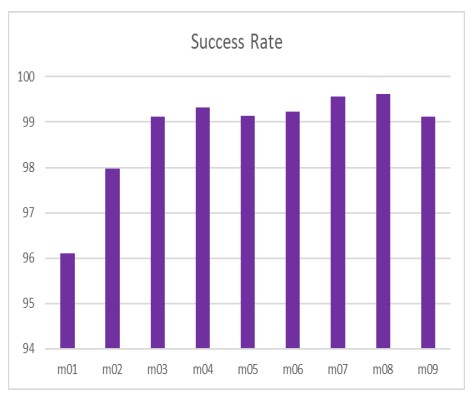

Figure 8: Success rate evaluations of MTRL over automatic reward generation.

Figure 7: Generated goal position for 9 manipulation tasks.

As a summary, LARG$^2$ demonstrates its capability of producing valid reward functions to successfully train and execute MTRL policies from textual task descriptions.

## 6 LIMITATIONS AND FUTURE WORKS

Our experiments have highlighted limitations in LLM reliability to convert user instructions into executable and valid code. Even though our experiments involved highly structured information such as function signature and docstring, which limits the effect of hallucination, the risk of semantic errors cannot be ruled out. To address these limitations, the auto-correction loop described in our paper seems an effective option to be further investigated.

## 7 CONCLUSION

In this paper, we introduce LARG$^2$ which enables scalable task-conditioned reinforcement learning from textual descriptions. Our method leverages the in-context learning and code-generation capabilities of large language models to complete or fully generate goal-sampling and reward functions from textual descriptions of tasks. For this purpose, our method incorporates automatic code validation and functional testing. Additionally, our approach augments the contextual information provided to the LLM with supplemental functions to activate a Chain-of-Thought mechanism to further increase the relevance of generated code. We evaluate the capability of our method to translate a series of text-based task descriptions into actionable objectives for GCRL and to generate rewards functions to train MTRL policies for robotic manipulation. Our experiment confirms the benefit of LARG$^2$ for aligning textual task descriptions with generated goal and reward functions. We believe it opens a novel and scalable direction for training RL-based policies for robots on the basis of textual instructions. Still, further work remains to address reward generation for long horizon objectives as well as for improvements in supplemental function retrieval using for instance a learning to rank approach.

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

# A    APPENDIX

In this section, we delve into further details regarding LARG$^2$ and present the results of experiments conducted to assess its performance in both Goal Conditioned Reinforcement Learning (GCRL) and Multi-Task Reinforcement Learning (MTRL) settings. Additionally, we analyze how the inclusion of supplementary function examples influences the quality and relevance of the generated code. Furthermore, we provide examples of the prompts used in our experiments.

## A.1    METHOD

### A.1.1    PREREQUISITES

LARG$^2$ offers a scalable approach for aligning language-based task descriptions with goal and reward functions, addressing both Goal-Conditioned and Multi-Task Reinforcement Learning challenges. This method harnesses the code generation capabilities provided by recent Large Language Models (LLMs) that encapsulate prior background knowledge and common sense. Regarding coding capabilities, these LLMs leverage existing code repositories like GitHub [12]. While one might argue that LLMs can generate appropriate code solely from textual descriptions, our experiments demonstrate that they still benefit from additional contextual guidelines. These guidelines encompass aspects such as scene understanding and function signature. Optionally, supplemental code samples can also be provided to activate a Chain-of-Thought (CoT) mechanism.

As a pre-requisite, we assume the existence of a set of categories of manipulation tasks defined in repositories like Isaac Gym [13] with descriptions of environments formalized using languages like YAML [14] or Python [15]. Accordingly, we assume that such environments provide signatures of expected functions commented with a formalism like Docstring [16].

In such a case, the search and retrieval process proposed in LARG$^2$ utilizes this information to align the generated code with the expected goal or reward function's signature. This facilitates seamless integration into existing training frameworks.

### A.1.2    GENERATION OF GOAL POSES FOR GCRL

A first application of LARG$^2$ generates goals to be used as parameters for goal-conditioned reward functions. As an example, within tabletop robotic manipulation scenarios, a pick-and-place task involves rearranging a set of objects that make up a scene. In this scenario, the goal is defined by a set of Cartesian coordinates representing target poses. Consequently, reward functions typically calculate the Euclidean distance between a given object and its corresponding target pose.

The prompt design used in LARG$^2$ is composed of 4 main elements: (1) the environment description, (2) the task description, (3) the specifications of the expected function and (4) optional guidelines. Optionally, it may be complemented by supplemental information like additional functions to initiate a Chain-of-Though (CoT) mechanism. The impact of such additional input is discussed in section A.1.4. Figure 9 illustrates our prompt design and figure 10 shows the resulting code.

---

[12]https://github.com/
[13]https://github.com/NVIDIA-Omniverse/IsaacGymEnvs
[14]https://yaml.org/
[15]https://www.python.org/
[16]https://peps.python.org/pep-0257/

```
We are implementing a table top rearrangement task within isaac gym such as Fra
nka_Move.
We need to set goal positions.

Could you complet the code of the python function "generate_goal_poses" below
with its body implementation
according to settings defined in the docstring below
for the goal: "Move the three cubes on the table so at the end they form a righ
t-angled triangle."

Code to be completed:

import torch
from torch import Tensor
from typing import Tuple

def generate_goal_poses() -> Tuple[Tensor, Tensor, Tensor]:
    """This function returns an eligible goal pose satisfying the following tab
le top rearrangement task: Move the three cubes on the table so at the end they
form a right-angled triangle."

    The setup is 1 table of dimension 1 x 1 x 0.79 on which stands 3 cubes of e
dge 5cm.
    Each cube has its own origin in its center.
    The global origin (0,0,0) is located on the floor below the middle of one o
f the table's sides.
    X coordinates range: (0.3, +0.90) = (Bottom, Top)
    Y coordinates range: (-0.4, +0.4) = (Right, Left)
    The minimum Z value for objects located on the surface of the table is 0.79
+ 0.05/2 = 0.815

    Returns:
     - Tuple: a tuple of 3 elements. Each element is a tensor . Each tensor of
shape (3) represents the goal position of one cube on the table
    """
```

Figure 9: GCRL: Prompt requesting the generation of the goal function. The function signature is highlighted in blue and the text-based goal description is in grey.

```
import torch
from torch import Tensor
from typing import Tuple

def generate_goal_poses() -> Tuple[Tensor, Tensor, Tensor]:
    """This function returns an eligible goal pose satisfying the following tab
le top rearrangement task: Move the three cubes on the table so at the end they
form a right-angled triangle."

    The setup is 1 table of dimension 1 x 1 x 0.79 on which stands 3 cubes of e
dge 5cm.
    Each cube has its own origin in its center.
    The global origin (0,0,0) is located on the floor below the middle of one o
f the table's sides.
    X coordinates range: (0.3, +0.90) = (Bottom, Top)
    Y coordinates range: (-0.4, +0.4) = (Right, Left)
    The minimum Z value for objects located on the surface of the table is 0.79
+ 0.05/2 = 0.815

    Returns:
     - Tuple: a tuple of 3 elements. Each element is a tensor . Each tensor of
shape (3) represents the goal position of one cube on the table
    """

    # Define the position of the cubes to form a right-angled triangle
    cube1_pos = torch.tensor([0.6, -0.2, 0.815])
    cube2_pos = torch.tensor([0.9, -0.2, 0.815])
    cube3_pos = torch.tensor([0.9, 0.1, 0.815])

    return cube1_pos, cube2_pos, cube3_pos
```

Figure 10: GCRL: Generated code for the goal pose function.

### A.1.3 GENERATION OF REWARD FUNCTION FOR MTRL

The second utilization of LARG$^2$ generates the executable source code of a reward function according to a task description.

For MTRL, the policy takes both the textual task description and the standard state input into account. In contrast to the GCRL case, goals are no longer explicitly present in the environment. However, this information remains crucial for the reward function, as it is required to compute a gain, which is generated in accordance with the provided task description.

To enhance efficiency and accommodate certain limitations in current Large Language Models (LLMs), we assume that the task reward is a composite sum of various components. We distinguish between task-independent components and task-dependent components. In the context of robotic manipulation, task-independant components typically include rewards for lifting objects or penalties for the number of actions required to reach the goal. Task-independent components can be sourced from existing code repositories or predefined environment settings, such as those in the Pick and Place scenario in Isaac Gym. LARG$^2$ is primarily focused on generating the portion of the reward that depends on specific requirements and constraints outlined in the guidelines and task descriptions.

The structure used for generating the reward function closely resembles the one employed for goal generation. It comprises (1) the environment description, (2) the task description, (3) specifications for the expected reward function, and (4) optional guidelines.

The following figures illustrate prompts and the results obtained when requesting the generation of a reward function to train a policy for manipulating a cube and bringing it closer to a robotic arm. For this example, we do not consider supplemental examples to condition code generation using Chain-of-Thought (CoT). Figure 11 details the global reward function that combines both elements from the task independent, which is illustrated by figure 12, and task dependent part. In this case, LARG$^2$ focuses on generating the dependent part using a prompt illustrated by Figure 13 to produce the code depicted in Figure 14.

```
def compute_franka_reward(
    reset_buf: Tensor, progress_buf: Tensor, successes: Tensor, actions: Tensor,
    lfinger_grasp_pos: Tensor, rfinger_grasp_pos: Tensor, object_pos: Tensor, goal_pos: Tensor,
    object_z_init: float, object_dist_reward_scale: float, lift_bonus_reward_scale: float,
    goal_dist_reward_scale: float, goal_bonus_reward_scale: float, action_penalty_scale: float,
    contact_forces: Tensor, arm_inds: Tensor, max_episode_length: int
) -> Tuple[Tensor, Tensor, Tensor]:

    # og_d: The distance between the object pose and the goal pose
    og_d = compute_object_to_goal_distance( object_pos, goal_pos)

    # object_above: Boolean, true if the object is above the table, false otherwise.
    object_above = is_object_above_initial_pose (object_pos, object_z_init)

    # Part of the reward that is task invariant
    static_rewards, reset_buf, lfo_dist_reward = compute_franka_reward_static( reset_buf, progress_
buf, successes, actions, lfinger_grasp_pos, rfinger_grasp_pos, object_pos, object_z_init, object_di
st_reward_scale, lift_bonus_reward_scale, goal_dist_reward_scale, goal_bonus_reward_scale, action_p
enalty_scale, contact_forces, arm_inds, max_episode_length)

    # Part of the reward that depends on the specifications provided in the task definition

    # og_d: The distance between the object pose and the goal pose
    og_d = compute_object_to_goal_distance( object_pos, goal_pos )
    # object_above: Boolean, true if the object is above the table, false otherwise.
    object_above = is_object_above_initial_pose (object_pos, object_z_init)

    # Compute generated part of the reward
    generated_rewards = compute_franka_reward_generated( lfo_dist_reward, object_above, og_d, goal_
dist_reward_scale, goal_bonus_reward_scale)

    # Total reward
    rewards = static_rewards \
            + generated_rewards

    # Goal reached
    successes = compute_successes(og_d, successes)

    return rewards, successes
```

Figure 11: MTRL: Source code of a global reward function combining a task independent and task dependent component (highlighted section in yellow).

```
def compute_franka_reward_static(
    reset_buf: Tensor, progress_buf: Tensor, successes: Tensor, actions: Tensor,
    lfinger_grasp_pos: Tensor, rfinger_grasp_pos: Tensor, object_pos: Tensor, goal_pos: Tensor,
    object_z_init: float, object_dist_reward_scale: float, lift_bonus_reward_scale: float,
    goal_dist_reward_scale: float, goal_bonus_reward_scale: float, action_penalty_scale: float,
    contact_forces: Tensor, arm_inds: Tensor, max_episode_length: int
) -> Tuple[Tensor, Tensor, float]:

    # Left finger to object distance
    lfo_d = torch.norm(object_pos - lfinger_grasp_pos, p=2, dim=-1)
    lfo_d = torch.clamp(lfo_d, min=0.02)
    lfo_dist_reward = 1.0 / (0.04 + lfo_d)

    # Right finger to object distance
    rfo_d = torch.norm(object_pos - rfinger_grasp_pos, p=2, dim=-1)
    rfo_d = torch.clamp(rfo_d, min=0.02)
    rfo_dist_reward = 1.0 / (0.04 + rfo_d)

    # Object above table
    object_above = (object_pos[:, 2] - object_z_init) > 0.015

    # Above the table bonus
    lift_bonus_reward = torch.zeros_like(lfo_dist_reward)
    lift_bonus_reward = torch.where(object_above, lift_bonus_reward + 0.5, lift_bonus_reward)

    # Regularization on the actions
    action_penalty = torch.sum(actions ** 2, dim=-1)

    # Total reward
    rewards = object_dist_reward_scale * lfo_dist_reward + object_dist_reward_scale * rfo_dist_reward
+ lift_bonus_reward_scale * lift_bonus_reward - action_penalty_scale * action_penalty

    # Object below table height
    object_below = (object_z_init - object_pos[:, 2]) > 0.04

    reset_buf = torch.where(object_below, torch.ones_like(reset_buf), reset_buf)

    # Arm collision
    arm_collisions = torch.any(torch.norm(contact_forces[:, arm_inds, :], dim=2) > 1.0, dim=1)

    reset_buf = torch.where(arm_collisions, torch.ones_like(reset_buf), reset_buf)

    # Max episode length exceeded
    reset_buf = torch.where(progress_buf >= max_episode_length - 1, torch.ones_like(reset_buf), reset_
buf)

return rewards, reset_buf, lfo_dist_reward
```

Figure 12: MTRL: Source code of the task independent reward component.

```
Context: We are developing in python a reward function for a Franka_move task in Isaac_gym.
This function returns a tuple composed of the reward for achieving the objective.
The objective is the following table top rearrangement task: "Take the cube and put it close to th
e robot arm."
This reward is composed of the object to goal reward and the bonus if object is near the goal

Complete this function, setting reward function to optimize the distance between the object and it
s goal pose .

import torch
from torch import Tensor
from typing import Tuple

from gpt.utils import *
from envs.utils.torch_jit_utils import *

@torch.jit.script
def compute_franka_reward_generated(
    lfo_dist_reward: float,
    object_above: bool,
    og_d: float,
    goal_dist_reward_scale: float,
    goal_bonus_reward_scale: float
) -> Tuple[Tensor]:
    """
    Compute the reward signal for a Franka Move task in Isaac Gym.
    Args:
        lfo_dist_reward: The reward for moving the left finger closer to the object.
        object_above: Boolean, true if the object is above the table, false otherwise.
        og_d: The distance between the object pose and the goal pose
        goal_dist_reward_scale: Scaling factor for the goal distance reward.
        goal_bonus_reward_scale: Scaling factor for the goal bonus reward.
    Returns:
        A tuple containing:
            - rewards: The reward signal for the current state.
    """

    # Add the code for computing the part of the reward related to optimizing the distance betwee
n the object and its goal pose

    return rewards
```

Figure 13: MTRL: Prompt requesting the generation of a task dependent part of a reward function.

```
import torch
from torch import Tensor
from typing import Tuple

def compute_franka_reward_generated(
    lfo_dist_reward: float, object_pos: Tensor, og_d: Tensor,
    object_z_init: float,
    goal_dist_reward_scale: float, goal_bonus_reward_scale: float
) -> Tuple[Tensor, float]:

    # Object above table
    object_above = (object_pos[:, 2] - object_z_init) > 0.015

    # Object to goal reward
    og_dist_reward = torch.zeros_like(lfo_dist_reward)
    og_dist_reward = torch.where(object_above, 1.0 / (0.04 + og_d), og_dist_reward
    )

    # Bonus if object is near the goal
    og_bonus_reward = torch.zeros_like(og_dist_reward)
    og_bonus_reward = torch.where(og_d <= 0.04, og_bonus_reward + 0.5, og_bonus_re
ward)

    # Total reward
    rewards = goal_dist_reward_scale * og_dist_reward  + goal_bonus_reward_scale *
og_bonus_reward

    return rewards
```

Figure 14: MTRL: Source code generated by LARG$^2$ for the task dependent part of a reward function.

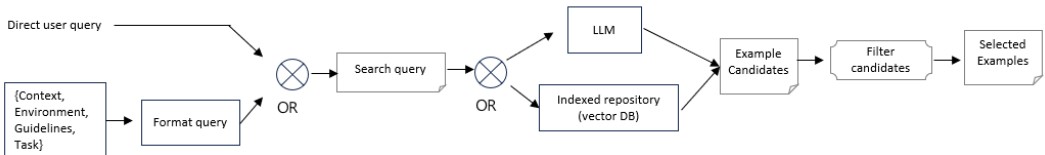

**Search & Retrieve pipeline**

Figure 15: The LARG[2] system enhanced with retrieval leverages context, environment, guidelines and task descriptions to query either the parametric memory of a LLM or a code database to retrieve function samples. These samples serve as additional context, enriching a dedicated prompt. This prompt is then utilized to convert textual task descriptions into goal poses or reward functions, a crucial process for applications like GCRL or MTRL.

### A.1.4 SEARCH AND RETRIEVAL FOR SUPPLEMENTARY EXAMPLES

The prompt can be enriched with function examples to guide intermediate reasoning steps. To achieve this, we can leverage either the contextual knowledge embedded within the parametric space of a large language model pre-trained on code repositories or employ a search and retrieval process from an external source of code examples.

Our method assumes the availability of code repositories such as Github[17]. To ensure the efficiency of the search and retrieval process, it is important for these repositories to contain sufficient documentation and comments. Additionally, while it is preferable for this code to be correct, previous research conducted by Wang et al. (2023a) has demonstrated that even invalid examples used in a Chain-of-Thought mechanism can still yield valid answers.

As depicted in Figure 15, we propose two viable options. Firstly, these repositories can be integrated into a dataset used for training a Large Language Model (LLM), in which case the information becomes embedded within the model's parametric memory, accessible for retrieval. Alternatively, repositories can be independently indexed, possibly utilizing a vector database that stores information as high-dimensional vectors. This latter option enables the augmentation of the LLM's background knowledge with external information sources, which may be more pertinent for specific applications.

While we do evaluate code retrieval from general-purpose Large Language Models (LLMs) in our experiments, our primary focus here centers on a dedicated external codebase search process. To achieve this, we implement a dense indexing system encompassing code, comments, and documentation extracted from code repositories.

In this indexing approach, each code file undergoes segmentation into a set of functions, with each function being individually indexed. This indexing process, denoted as $I$, aggregates information from various sources, including the readme.md file ($R$), the function's signature ($S$), its docstring ($D$), and its code ($C$). This aggregation can be represented as $R, S, D, C \rightarrow F$, where $F$ represents the indexed function. The result is encoded into a collection of embeddings and subsequently stored within a Vector database. As a result, each query to the vector database returns a sorted set of individual functions, ordered according to a relevance score. The expected number of relevant functions is provided as a parameter of the query.

### A.1.5 CODE VALIDATION

To ensure the generated code seamlessly integrates with existing reinforcement learning frameworks, it may be necessary to perform code validation followed by potential code correction. To accomplish this, we harness the code generation capabilities of Large Language Models (LLMs) to produce a functional test.

---

[17]https://www.github.com

We employ a structured prompt, as depicted in Figure 16, comprising the following components: (1) a header requesting the LLM to generate a functional test, (2) a list of guidelines to shape the test, and (3) the code for the generated function. An example of such a test is provided in Figure 17.

```
We are implementing a reward function of a custom task for a Franka Move environment within Isaac gym.
Our setting is: a table holding one Franka Emika robot arm and 3 cubes of edge 5cm.
The dimensions of the table are: (1m x 1m x 0.78m).
The robot base position is in the middle of one of the table's sides at the 3D coordinates
(0.5, 0.165, 0.78m).
There is a gripper at the very end of the robot arm.

Our custom task is: "Move a cube to the top right corner of the table."

Update the following python script with a functional test for the reward function "compute_franka_reward"
Reward tests should only validate cases when they should be positive (>=0) or negative (<=0).
Success should be tested against 1 or 0 values.

def compute_franka_reward(object_pos: Tensor, lfinger_grasp_pos: Tensor, rfinger_grasp_pos: Tensor) -> Tu
ple[Tensor, Tensor]:
    """Our setting is: a table, a one arm robot, and 3 cubes of edge 5cm.
                                            …
```

Figure 16: Prompt requesting the generation of functional test for a reward function.

```
# Functional tests for the compute_franka_reward function
def test_compute_franka_reward():
    # Test case 1: Object positions at target position, gripper positions at target position
    object_pos = torch.tensor([[[0.95, 0.95, 0.15], [0.9, 0.9, 0.15], [0.8, 0.8, 0.15]]])
    lfinger_grasp_pos = torch.tensor([[0.95, 0.95, 0.15]])
    rfinger_grasp_pos = torch.tensor([[0.95, 0.95, 0.15]])
    expected_rewards = torch.tensor([[0.0, 0.0, 0.0]])
    expected_successes = torch.tensor([[1.0, 1.0, 1.0]])
    rewards, successes = compute_franka_reward(object_pos, lfinger_grasp_pos, rfinger_grasp_pos)
    assert torch.allclose(rewards, expected_rewards)
    assert torch.allclose(successes, expected_successes)

    # Test case 2: Object positions far from target position, gripper positions far from target position
    object_pos = torch.tensor([[[0.5, 0.5, 0.15], [0.6, 0.6, 0.15], [0.7, 0.7, 0.15]]])
    lfinger_grasp_pos = torch.tensor([[0.5, 0.5, 0.15]])
    rfinger_grasp_pos = torch.tensor([[0.5, 0.5, 0.15]])
    expected_rewards = torch.tensor([[-0.4871, -0.5831, -0.7031]])
    expected_successes = torch.tensor([[0.0, 0.0, 0.0]])
    rewards, successes = compute_franka_reward(object_pos, lfinger_grasp_pos, rfinger_grasp_pos)
    assert torch.allclose(rewards, expected_rewards)
    assert torch.allclose(successes, expected_successes)

    # Test case 3: Object positions close to target position, gripper positions close to target position
    object_pos = torch.tensor([[[0.95, 0.95, 0.25], [0.9, 0.9, 0.25], [0.8, 0.8, 0.25]]])
    lfinger_grasp_pos = torch.tensor([[0.95, 0.95, 0.25]])
    rfinger_grasp_pos = torch.tensor([[0.95, 0.95, 0.25]])
    expected_rewards = torch.tensor([[0.15, 0.15, 0.15]])
    expected_successes = torch.tensor([[1.0, 1.0, 1.0]])
    rewards, successes = compute_franka_reward(object_pos, lfinger_grasp_pos, rfinger_grasp_pos)
    assert torch.allclose(rewards, expected_rewards)
    assert torch.allclose(successes, expected_successes)

    print("All tests passed!")
```

Figure 17: Generated functional test.

## A.2 EXPERIMENTS

We evaluate LARG[2] on a series of tabletop object manipulation tasks for both GCRL and MTRL settings. First, we focus on evaluating goal and reward generation without supplemental examples. Then, we evaluate the benefits of a chain-of-thought mechanism. Experiments leverage the Franka_Move environment available on the IsaacGym repository [18]. This environment incorporates a table, a Franka Emika Panda robot arm [19] which is an open kinematic chain composed with 7DoF, and $n$ cubes on the table. The dimensions of the table are as follows: 1m x 1m x 0.78m. The robot arm is placed on the table at (0.5, 0.165, 0.78). There is a griper with two fingers attached at the end of the arm. Cubes with a 5cm edge are located on the surface of the table. The global origin (0,0,0) is located on the floor below the table. Each environment description is written using the Python language.

### A.2.1 LARGE LANGUAGE MODELS

In a first experimental setting, several LLMs are evaluated on a first list of 32 tasks: text-davinci-003 [20], code-davinci-002 [21] and gpt-3.5-turbo [22] from OpenAI which are evolutions from GPT3 optimized with Reinforcement Learning from Human Feedback Ouyang et al. (2022). StarCoder from HuggingFace (Li et al., 2023) is also evaluated on the same tasks but it only delivers 12.5% of valid functions even after the auto-correction step. Frequent issues observed with this model are related to incorrect variable initialization, missing code and a lack of compliance with provided guidelines such as illustrated in Figures 18 and 19.

```python
import torch
from torch import Tensor
from typing import Tuple
def generate_goal_pose() -> Tensor:
    """This function returns an elligible goal pose satisfying the following table top rearrangement
task: "move a cube in the top right corner of the table."

    The setup is 1 table of dimension 0.81 x 1 x 0.795 on which stands 1 cube of edge 5cm and ha
ving its origin in its center.
    The global origin (0,0,0) is located on the floor below the middle of one of the table's sides.
    X coordinates range: (0., +0.70) = (Bottom, Top)
    Y coordinates range: (-0.4, +0.4) = (Right, Left)
    The minimum Z value for objects located on the surface of the table is 0.795 + 0.05/2 = 0.82

    Returns:
     - Tensor: a tensor of shape (3) representing the goal position
    """
    x = 0.65
    y = -0.35
    z = 0.82
    goal_pose = torch.tensor([x, y, z])
    return goal_pose
```

Figure 18: Code generated by gpt-3.5-turbo for the task: Move a cube in the top right corner of the table.

As a summary, best results, in terms of code generation, are achieved with gpt-3.5-turbo which is the model used in a first series of experiments reported below.

---

[18]https://developer.nvidia.com/isaac-gym

[19]https://www.franka.de/

[20]https://platform.openai.com/docs/models/gpt-3

[21]https://platform.openai.com/docs/models/codex

[22]https://platform.openai.com/docs/models/gpt-3-5

```
import random
import numpy as np
import torch

def generate_goal_pose():
    # Define limits based on your environment dimensions
    max_x = <max-value-for-x>
    min_x = <min-value-for-x>
    max_y = <max-value-for-y>
    min_y = <min-value-for-y>
    max_z = <max-value-for-z>
    min_z = <min-value-for-z>
    while True:
        # Generate random values within defined ranges
        rand_vals = [random.uniform(min_x, max_x),
                     random.uniform(min_y, max_y),
                     random.uniform(min_z, max_z)]
        if check_valid(*rand_vals):
            break
    return torch.tensor([*rand_vals])

def check_valid(x, y, z):
    # Check validity conditions here, e.g.:
    # Does not collide with other objects?
    # Is within workspace boundaries?
    pass
if __name__ == "__main__":
    print("Example usage:")
    pos = generate_goal_pose().numpy()
    print(f"Generated Position: {pos}")
```

Figure 19: Code generated by StarCoder for the task: Move a cube in the top right corner of the table. In this example the generated code cannot be applied.

### A.2.2 AUTOMATIC GOAL GENERATION FOR THE GCRL EXPERIMENT

In the GCRL experiment, the policy takes as input the position and velocity of each joint of the robot and the respective pose of the objects composing the scene. The policy triggers joint displacement in a $\mathbb{R}^7$ action space. In addition to the position of the object composing the scene, the policy takes as input the goal positions. These positions are provided by goal functions generated by LARG[2]. The policy is trained beforehand using Proximal Policy Optimization Schulman et al. (2017) with default Franka_Move parameters as defined in table 3.

| training parameters | values |
|---|---|
| number of environments | 2048 |
| episode length | 500 |
| object distance reward scale | 0.08 |
| lift bonus reward scale | 4.0 |
| goal distance reward scale | 1.28 |
| goal bonus reward scale | 4.0 |
| action penalty scale | 0.01 |
| collision penalty scale | 1.28 |
| actor hidden dimension | [256, 128, 64] |
| critic hidden dimension | [256, 128, 64] |

Table 3: List of parameters used in the Franka_Move PPO training loop.

We assess our approach across an initial set of 32 tasks, consisting of 27 tasks that involve a single object and 5 tasks that encompass three objects. Tasks labeled $d17$ to $d27$ are characterized by objectives defined in relation to the initial positions of the objects. In such cases, the goal function's signature naturally incorporates the initial positions of the cubes comprising the scene.

Figure 20 illustrates the workflow for generating prompts that transform task descriptions into goal function generation. This process includes an auto-correction step and the subsequent creation of a functional test. In this experiment, we refrain from using supplemental functions to enhance the prompt with additional context.

Figure 21 illustrates the results produced with 10 runs of 3 different goal functions generated out of 3 different manipulation tasks. In all cases, the resulting poses are well aligned with task requirements while exploring the range of valid positions allowed by a non deterministic task definition.

Table 4 provides the list of all tasks used in our experiment and report the compliance of generated goals with task descriptions.

| ID | Task | Generated Pose validity |
|---|---|---|
| d01 | Move a cube to the top right corner of the table. | ✓ |
| d02 | Move a cube to the top left corner of the table. | ✓ |
| d03 | Move a cube to the bottom right corner of the table. | ✓ |
| d04 | Move a cube to the bottom left corner of the table. | ✓ |
| d05 | Lift the cube 15cm above the table. | ✓ |
| d06 | Rotate a cube upside-down. | ✓ |
| d07 | Take to cube and move it to the left side of the table. | - |
| d08 | Take to cube and move it to the right edge of the table. | ✓ |
| d09 | Take to cube and raise it at 20 cm to the far side of the table. | ✓ |
| d10 | Take the cube and move it closer to the robotic arm. | ✓ |
| d11 | Pick up the cube and move it away from the robotic arm. | ✓ |
| d12 | Take the cube and move it very close to the robotic arm. | - |
| d13 | Push the cube off the limits of the table. | ✓ |
| d14 | Bring the cube closer to the robot arm. | ✓ |
| d15 | Move the cube to one corner of the table. | ✓ |
| d16 | Place the cube anywhere on the diagonal of the table running from the top right corner to the bottom left corner. | ✓ |
| d17 | Lift the cube 15cm above the table and 10 cm to the right. | ✓ |
| d18 | Lift the cube 20cm above the table and 15 cm ahead. | ✓ |
| d19 | Lift the cube 20cm above the table and 15 cm backward. | ✓ |
| d20 | Push a cube 10cm to the right and 10cm ahead. | ✓ |
| d21 | Push a cube 10cm to the right and 10cm backward. | ✓ |
| d22 | Push a cube 10cm to the left and 10cm ahead. | ✓ |
| d23 | Push a cube 10cm to the left and 10cm backward | ✓ |
| d24 | Grab a cube and move it a bit to the left. | ✓ |
| d25 | Grab a cube and lift it a bit and move it a bit ahead. | ✓ |
| d26 | Move the cube at 20cm to the left of its initial position. | ✓ |
| d27 | Move the cube 20cm above its current position. | ✓ |
| d28 | Move one cube to the left side of the table, another one to the right side of the table, and put the last cube at the center of the table. | ✓ |
| d29 | Move the three cubes so they are 10 cm close to one another. | ✓ |
| d30 | Move the three cubes on the table so that at the end they form a right-angled triangle. | ✓ |
| d31 | Move the three cubes on the table so that at the end they form an isosceles triangle. | ✓ |
| d32 | Reposition the three cubes on the table such that they create a square, with the table's center serving as one of the square's corners. | ✓ |

Table 4: List of the 32 manipulation tasks evaluated with $LARG^2$. Tasks $d17$ to $d27$ involve objectives relative to the object's initial position. Tasks $d28$ to $d32$ address 3 object manipulation problems and therefore 3 goals. Localisation compliance with task definition is reported.

Figure 22 displays the success rates for our set of 32 manipulation tasks. When examining unsuccessful experiments, a frequent source of error can be traced back to insufficient contextual information and constraints within the task definition.

As potential future directions to tackle this issue, two options can be mentioned: firstly, enhancing the prompt with more constraints, and secondly, opting for a more capable Large Language Model (LLM) with respect to code generation capabilities.

Interestingly, this experiment underscores the reasoning capabilities of the Large Language Model (LLM), as depicted in Figure 23. In this specific task, the requirement is to lift a cube to a height of 15cm above the table. Remarkably, the generated goal function demonstrates the ability to correctly calculate the target position by adding the table's height to the specified 15cm.

As a summary, $LARG^2$ allows to generate code for goal prediction according to textual task descriptions. In some cases, the generated code do not properly fits with user specifications but our experiment demonstrates that a feedback loop with additional guidelines can fix the problem.

```
We are implementing a table top rearrangement task within Isaac gym.
We need to set goal positions.

Could you complete the code of the python function "generate_goal_pose" below
with its body implementation according to settings defined in the docstring below
for the goal: "Move the three cubes on the table so at the end they form a right-a
ngled triangle."

Code to be completed:

import torch
from torch import Tensor
from typing import Tuple

def generate_goal_pose() -> Tuple[Tensor]:
    """"This function returns an eligible goal pose satisfying the following tablet
op rearrangement task: "nest a cube in the top right corner of the table"

    The setup is 1 table of dimension 0.81 x 1 x 0.795 on which stands 1 cube of e
dge 5cm and having its origin in its center.
    The global origin (0,0,0) is located on the floor below the middle of one of t
he table's sides.
    X coordinates range: (0., 0.70)
    Y coordinates range: (-0.4, 0.4)
    The minimum Z value for objects located on the surface of the table is 0.795 +
0.05/2 = 0.82

    Returns :
     - Tensor: a tensor of shape (3) representing the goal position
    """

It is important to leave the function signature unchanged and keep the docstring a
s is.
Do not generate sample usage nor inner functions.
Double-check for any unused variables or functions, missing or incorrect imports,
punctuation marks, indentation errors, or mismatched parentheses/brackets.
```

```
Could you please fix the error:
'AssertionError: <function generate_goal_pose at 0x7f4bec4bf550> should return one
element of shape 3: (tensor([0.5821, 0.1927, 0.8200]),)'
in the following function implementation:

import torch
from torch import Tensor
from typing import Tuple
import random

def generate_goal_pose() -> Tuple[Tensor]:
    """
    This function returns an eligible goal pose satisfying the following tabletop
rearrangement task: "nest a cube in the top right corner of the table."

    The setup is 1 table of dimension 0.81 x 1 x 0.795 on which stands 1 cube of e
dge 5cm and having its origin in its center.
    The global origin (0, 0, 0) is located on the floor below the middle of one of
the table's sides.
    X coordinates range: (0., 0.70)
    Y coordinates range: (-0.4, 0.4)
    The minimum Z value for objects located on the surface of the table is 0.795 +
0.05/2 = 0.82

    Returns:
        - Tensor: a tensor of shape (3) representing the goal position
    """
    x = random.uniform(0, 0.7)
    y = random.uniform(0, 0.4)
    z = 0.82
    return (torch.tensor([x, y, z]),)
```

```
Update the following python script with functional tests for the goal position
function 'generate_goal_pose'.
Do no add any explanation text.
Return the same script plus what you have inserted.

#Functional Tests
def test_generate_goal_pose():
    print("starting functional tests")

    #
    # insert the functional tests here
    #
```

Figure 20: Prompts illustrating the 3 steps involved in the generation of a valid goal positioning function: 1) request to generate a function according to specific environment parameters, 2) auto-correction, 3) final validation. The highlighted section in red contains the error message generated at the execution phase.

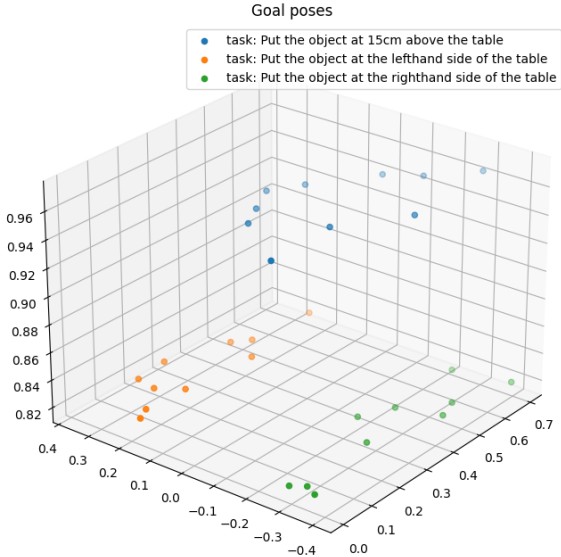

Figure 21: Example of goal positions generated by our method for 3 different tasks requesting targets to be located on the right, left, and above the table.

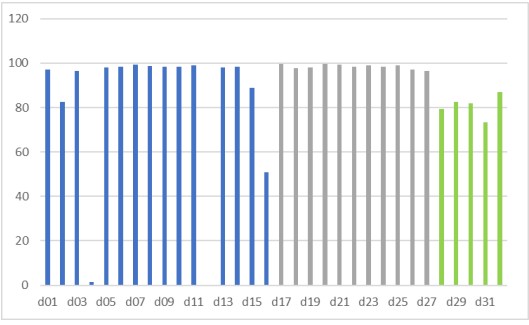

Figure 22: Success rate for GCRL manipulation tasks. Blue reflects 1 object manipulation for absolute pose whereas grey reflects relative object pose. Green relates to 3 object manipulation tasks.

```python
import torch
from torch import Tensor
from typing import Tuple

def generate_goal_pose() -> Tensor:
    # Define the minimum and maximum allowed positions
    min_x, max_x = 0.0, 0.7
    min_y, max_y = -0.4, 0.4
    min_z = 0.82
    target_z = min_z + 0.15   # Add 15 cm to the minimum Z value

    # Generate random X and Y coordinates within the allowed ranges
    x = torch.rand(1) * (max_x - min_x) + min_x
    y = torch.rand(1) * (max_y - min_y) + min_y

    # Create a tensor with the goal position
    goal_position = torch.tensor([x, y, target_z])

    return goal_position
```

Figure 23: Arithmetic capabilities of the LLM for Task *d05*. The comment highlighted in yellow so as the related code is generated by the LLM.

### A.2.3 AUTOMATIC REWARD GENERATION FOR THE MTRL EXPERIMENT

Our second experiment evaluates LARG$^2$ capabilities to address MTRL settings. For task encoding, we use the Google T5-small language model Raffel et al.. We use the *[CLS]* token embedding computed by the encoder stack of the model which is defined in $\mathbb{R}^{512}$. We concatenate this embedding with the state information of our manipulation environment defined in $\mathbb{R}^7$ and feed it into a fully connected layer stack used as policy. This policy is composed of three layers using respectively, $\{512, 128, 64\}$ hidden dimensions.

In our experiment, we train an MTRL settings using Proximal Policy Optimization (PPO) Schulman et al. (2017) with default Franka Move parameters using reward functions generated by LARG$^2$ over 9 tasks listed in Table 5. These tasks address one object manipulation on a tabletop. We leverage the LLM capabilities to paraphrase these tasks to produce the evaluation set. Paraphrases include task translation as the Google T5 model is trained for downstream tasks such as machine translation. Figure 25 illustrates the application of Task $m04$ submitted as a text based command in Korean language ("큐브를 테이블 중앙으로부터 20cm 위로 옮겨주세요") to a policy trained in MTRL.

| ID | Task |
|----|------|
| m01 | Push the cube to the far right of the table. |
| m02 | Move a cube to the top left corner of the table. |
| m03 | Take the cube and put it close to the robot arm. |
| m04 | Move a cube at 20cm above the center of the table. |
| m05 | Move a cube at 15 cm above the table. |
| m06 | Take the cube and put it on the diagonal of the table. |
| m07 | Push the cube at 20cm ahead of its current position. |
| m08 | Move the cube to the center of the table. |
| m09 | Grab the cube and move it forward to the left. |

Table 5: List of task used in the MTRL settings.

Figure 24 provides success rates obtained for the 9 tasks used in the MTRL experiment. It illustrates LARG$^2$ capabilities to generate valid reward functions to train and execute MTRL policies conditioned by textual task definitions.

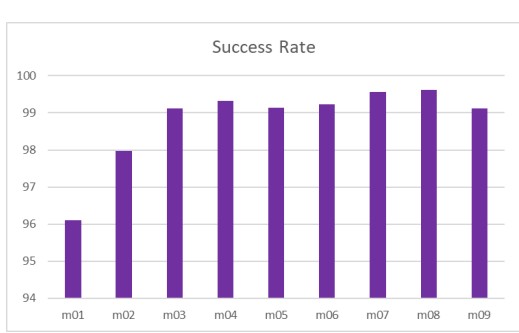

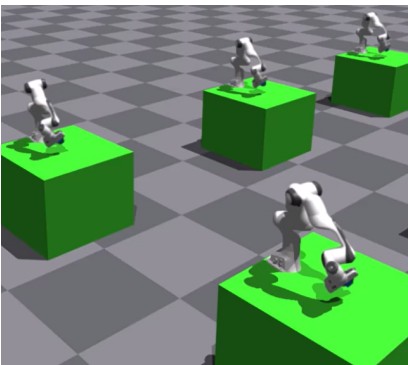

Figure 24: Success rate evaluations of MTRL over automatic reward generation.

Figure 25: Example of multi-lingual capabilities for robot manipulation. In our simulation tasks are submitted using different languages including English, Arabic and Korean. This figure illustrate task $m04$ translated in Korean.

### A.2.4 INFLUENCE OF SUPPLEMENTAL EXAMPLES FOR GENERATED GOAL RELEVANCE

Here, we evaluate the benefits of supplemental examples added to the main prompt on the relevance of generated goals. For this purpose, we design two experiments. The first experiment uses examples generated by a large language model, in others words , retrieved from the parametric space. The second experiment uses a search and retrieval mechanism leveraging an external code base.

**Leveraging the parametric space memory of a LLM**

For this experiment, we extract support functions from the parametric space of Large Language Models, leveraging their ability to generate code based on textual queries. In other words, we query the LLM itself to generate support functions.

We use three different queries to generate supplemental functions to be added to our main prompt where each example relates to a different function.:

- "Create a function to draw a bounding box within min max coordinates."
- "Create a function to test if an object is located within a bounding box."
- "Create a function to position randomly an object inside a given bounding box."

In this evaluation, we use three LLMs: GPT4 (GPT) [23], Hyper Clova X (HCX) [24], and StarCoder (SC)[25]. These models are used in the LARG$^2$ pipeline either in a straightforward manner without supplemental examples in the prompt, or with retrieved functions (RF) provided as examples.

| Task | GPT | GPT+RF | HCX | HCX+RF | SC | SC+RF |
|---|---|---|---|---|---|---|
| Move a cube in the top right corner of the table. | **0.8** | 0.75 | 1 | 0.25 | 0 | 0 |
| Lift the cube 15cm above the table. | 0.8 | **0.9** | 0.8 | 0 | 0 | 0 |
| Take the cube and move it to the left side of the table. | **1** | 0.75 | **1** | 0.5 | 0.25 | 0 |
| Take the cube and move it closer to the robotic arm. | 0.4 | 0.5 | 0 | **1** | 0 | 0 |
| Move the cube 20cm to the left of its initial position. | 0.5 | **0.75** | 0.5 | 0 | 0 | 0 |

Table 6: Performance comparison between three LLMs: GPT4 (GPT), HyperClovaX (HCX), and StarCoder (SC).

Inspired by the results presented in table 6 which highlight a positive impact of supplemental examples using the GPT4 model, we then use this model to test a search and retrieval approach from an external code base.

**Search and retrieval from an external: code repository**

To assess the ability to automatically retrieve pertinent examples from code repositories using automated queries, we start by indexing and storing these repositories within a vector database.

In our experiment, we use The Stack [26] as a code base. The Stack contains over 6TB of permissively-licensed source code files covering 358 programming languages build as part of the BigCode project [27] from repositories like those available in GithHub [28].

For the sake of performance, we filter this dataset, retaining only Python files from repositories related to robot learning for manipulation tasks. We use text-based information found in markdown files associated with each repository for this filtering process.

Once filtered, we index and store this dataset in a vector database. In our experiment, we use ChromaDB [29]. Repository descriptions, comments and function names are encoded using Sentence-Transformer [30]. This encoding served a dual purpose: to create the embedding vector used to retrieve relevant functions and to encode queries. This indexed dataset serves as our external memory for retrieving examples to enrich the primary prompt in LARG$^2$ .

---

[23]https://openai.com/gpt-4

[24]https://clova.ai/hyperclova

[25]https://huggingface.co/blog/starcoder

[26]https://huggingface.co/datasets/bigcode/the-stack

[27]https://www.bigcode-project.org/

[28]https://github.com/

[29]https://www.trychroma.com/

[30]https://www.sbert.net/

To assess the effectiveness of the search and retrieval process, we examine the impact of two parameters: the number of examples provided as support functions and the alignment, or lack thereof, between the names of these functions and the name of the expected function. Table 7 presents a list of the combinations explored in our experiment. Specifically, we include from one to three functions as examples. Furthermore, we either select the function with the highest relevance score or conduct a random draw from the top four functions. Lastly, we experiment with modifying the support function names to match the expected name of function. This particular modification is inspired by an observation provided in Wang et al. (2023a), which underscores the importance of name coherence within the Chain-of-Thought mechanism.

| Prompt Id | Prompt Content |
|---|---|
| I_2 | Use the 2 top-ranked functions. Function names are not modified. |
| I_3 | Use the 3 top-ranked functions. Function names are not modified. |
| I_b | Top-ranked function. Its name is not modified. |
| I_r | A random function among the top 4 without considering the best one. The name is not modified. |
| L | LARG$^2$ without supplemental functions. |
| M_2 | Same as I_2 but function names are modified to match the targeted function signature. |
| M_3 | Same as I_3 but function names are modified to match the targeted function signature |
| M_b | Same as I_b but function names are modified to match the targeted function signature |
| M_r | Same as I_r but function names are modified to match the targeted function signature |

Table 7: Description of the diverse retrieved examples added the LARG$^2$ prompt to request code generation.

Table 8 compares the validity of generated goal poses with respect to textual task descriptions.

| Task | L | I_b | I_r | I_2 | I_3 | M_b | M_r | M_2 | M_3 |
|---|---|---|---|---|---|---|---|---|---|
| Move a cube in the top right corner of the table. | 0.75 | 0.7 | **0.9** | **0.9** | 0.4 | 0.5 | **0.9** | **0.9** | 0.25 |
| Lift the cube 15cm above the table. | **1.0** | **1.0** | 0.8 | 0.9 | 0.8 | 0.0 | **1.0** | **1.0** | **1.0** |
| Take the cube and move it to the left side of the table. | **1.0** | **1.0** | **1.0** | **1.0** | **1.0** | **1.0** | 0.3 | 0.6 | **1.0** |
| Take the cube and move it closer to the robotic arm. | 0.5 | 0.6 | 0.6 | 0.4 | 0.3 | 0.3 | **0.7** | 0.3 | **0.7** |
| Lift the cube 20cm above the table and 15 cm ahead. | 0.5 | **1.0** | **1.0** | **1.0** | **1.0** | **1.0** | **1.0** | **1.0** | **1.0** |
| Push a cube 10cm to the right and 10cm backward. | 0.2 | 0.5 | 0.2 | 0.2 | 0.5 | **0.6** | 0.5 | 0.5 | 0.5 |
| Grab a cube and lift it a bit and move it a bit ahead. | **1.0** | 0.5 | 0.7 | 0.8 | 0.7 | 0.7 | 0.7 | 0.8 | 0.9 |
| Move the cube at 20cm to the left of its initial position. | 0.5 | 0.6 | 0.7 | 0.8 | **0.9** | 0.7 | 0.5 | 0.5 | 0.5 |
| Move one cube to the left side of the table, another one to the right side of the table, and put the last cube at the center of the table. | 0.9 | 0.7 | 0.8 | 0.5 | 0.4 | **1.0** | 0.6 | 0.7 | 0.7 |
| Move the three cubes so they are 10 cm close to one another. | 0.9 | **1.0** | **1.0** | 0.3 | 0.2 | **1.0** | **1.0** | 0.2 | 0.2 |
| Move the three cubes on the table so that at the end they form a right-angled triangle. | **1.0** | **1.0** | 0.9 | 0.2 | 0.1 | **1.0** | **1.0** | 0.3 | 0.2 |
| Reposition the three cubes on the table such that they create a square, with the table's center serving as one of the square's corners. | 0.8 | 0.2 | **0.9** | 0.2 | 0.1 | 0.8 | **0.9** | **0.9** | 0.8 |

Table 8: Evaluation of LARG$^2$ performance for goal pose generation according to various configurations of the code example part of the prompt.

This experiment showcases two key aspects: the ability of LARG$^2$ to generate goals that align with the requirements outlined in task descriptions, and the impact of including supplementary code samples on the precision of the generated functions. Moreover, it underscores that the naming of functions has a minimal effect on performance.

## A.3 Examples of goal functions generated by LARG[2]

```python
import torch
from torch import Tensor
from typing import Tuple

def generate_goal_pose() -> Tensor:
    """This function returns an elligible goal pose satisfying the following table top rearrangement task: "can you take to cube and move it to the
right edge of the table."

    The setup is 1 table of dimension 0.81 x 1 x 0.795 on which stands 1 cube of edge 5cm and having its origin in its center.
    The global origin (0,0,0) is located on the floor below the middle of one of the table's sides.
    X coordinates range: (0., +0.70) = (Bottom, Top)
    Y coordinates range: (-0.4, +0.4) = (Right, Left)
    The minimum Z value for objects located on the surface of the table is 0.795 + 0.05/2 = 0.82

    Returns:
     - Tensor: a tensor of shape (3) representing the goal position
    """
    # The goal position will be at the right edge of the table, so we can set the y-coordinate to the maximum value of the range
    goal_pose = torch.tensor([0.35, 0.4, 0.82])
    return goal_pose
```

Figure 26: Task $d08$: Take to cube and move it to the right edge of the table

```python
import torch
from torch import Tensor
from typing import Tuple

def generate_goal_pose() -> Tensor:
    """This function returns an elligible goal pose satisfying the following table top rearrangement task: "take the cube and move it very close to the
robotic arm."

    The setup is 1 table of dimension 0.81 x 1 x 0.795 on which stands 1 cube of edge 5cm and having its origin in its center.
    The global origin (0,0,0) is located on the floor below the middle of one of the table's sides.
    X coordinates range: (0., +0.70) = (Bottom, Top)
    Y coordinates range: (-0.4, +0.4) = (Right, Left)
    The minimum Z value for objects located on the surface of the table is 0.795 + 0.05/2 = 0.82

    Returns:
     - Tensor: a tensor of shape (3) representing the goal position
    """
    # move the cube very close to the robotic arm (at y=-0.35) and 5cm above the table surface
    goal_position = torch.tensor([0.6, -0.35, 0.82])
    return goal_position
```

Figure 27: Task $d12$: Take the cube and move it very close to the robotic arm.

```python
import torch
from torch import Tensor
from typing import Tuple

def generate_goal_pose() -> Tensor:
    """This function returns an elligible goal pose satisfying the following table top rearrangement task: "move the cube to one corner
of the table."

    The setup is 1 table of dimension 0.81 x 1 x 0.795 on which stands 1 cube of edge 5cm and having its origin in its center.
    The global origin (0,0,0) is located on the floor below the middle of one of the table's sides.
    X coordinates range: (0., +0.70) = (Bottom, Top)
    Y coordinates range: (-0.4, +0.4) = (Right, Left)
    The minimum Z value for objects located on the surface of the table is 0.795 + 0.05/2 = 0.82

    Returns:
     - Tensor: a tensor of shape (3) representing the goal position
    """
    goal_pose = torch.tensor([0.7, -0.4, 0.82])
    return goal_pose
```

Figure 28: Task $d15$: Move the cube to one corner of the table.

```
import torch
from torch import Tensor
from typing import Tuple

def generate_goal_pose(cube: Tensor) -> Tensor:
    """
    This function returns an elligible goal pose satisfying the following table top rearrangement task: "place the cube anywhere on the
diagonal of the table  running from the top right corner to the bottom left corner."

    The setup is 1 table of dimension 0.81 x 1 x 0.795 on which stands 1 cube of edge 5cm and having its origin in its center.
    The global origin (0,0,0) is located on the floor below the middle of one of the table's sides.
    X coordinates range: (0., +0.70) = (Bottom, Top)
    Y coordinates range: (-0.4, +0.4) = (Right, Left)
    The minimum Z value for objects located on the surface of the table is 0.795 + 0.05/2 = 0.82

    Args:
     - cube: a tensor of shape (3) representing the cube position on the table

    Returns:
     - a tensor of shape (3) representing the goal position on the table
    """
    goal_x = torch.tensor([0.7]).to(cube.device)
    goal_y = torch.tensor([-0.4]).to(cube.device)
    goal_z = torch.tensor([0.82]).to(cube.device)

    # calculate the slope of the diagonal line
    slope = -(goal_y / goal_x)

    # calculate the y intercept of the diagonal line
    y_int = goal_y - (slope * goal_x)

    # use the slope and y intercept to calculate the desired y position of the cube
    goal_y_cub = (slope * cube[0]) + y_int

    # create the goal pose tensor
    goal_pose = torch.tensor([cube[0], goal_y_cub, goal_z]).to(cube.device)

    return goal_pose
```

Figure 29: Task $d16$: Place the cube anywhere on the diagonal of the table running from the top right corner to the bottom left corner.

```
import torch
from torch import Tensor
from typing import Tuple

def generate_goal_pose() -> Tensor:
    """This function returns an eligible goal pose satisfying the following table top rearrangement task: "lift the cube 15cm above the
table and 10 cm to the right."

    The setup is 1 table of dimension 0.81 x 1 x 0.795 on which stands 1 cube of edge 5cm and having its origin in its center.
    The global origin (0,0,0) is located on the floor below the middle of one of the table's sides.
    X coordinates range: (0., +0.70) = (Bottom, Top)
    Y coordinates range: (-0.4, +0.4) = (Right, Left)
    The minimum Z value for objects located on the surface of the table is 0.795 + 0.05/2 = 0.82

    Returns:
     - Tensor: a tensor of shape (3) representing the goal position
    """
    goal_pose = torch.Tensor([0.35, 0.1, 0.82+0.15])
    return goal_pose
```

Figure 30: Task $d17$: Lift the cube 15cm above the table and 10 cm to the right.

```
import torch
from torch import Tensor
from typing import Tuple

def generate_goal_pose() -> Tensor:
    """This function returns an eligible goal pose satisfying the following table top rearrangement task: "lift the cube 20cm above the
table and 15 cm backward."

    The setup is 1 table of dimension 0.81 x 1 x 0.795 on which stands 1 cube of edge 5cm and having its origin in its center.
    The global origin (0,0,0) is located on the floor below the middle of one of the table's sides.
    X coordinates range: (0., +0.70) = (Bottom, Top)
    Y coordinates range: (-0.4, +0.4) = (Right, Left)
    The minimum Z value for objects located on the surface of the table is 0.795 + 0.05/2 = 0.82

    Returns:
     - Tensor: a tensor of shape (3) representing the goal position
    """

    # We need to lift the cube vertically by 20 cm and move it back by 15 cm
    x_pos = 0.7
    y_pos = -0.15
    z_pos = 0.82 + 0.05

    goal_pose = torch.tensor([x_pos, y_pos, z_pos])

    return goal_pose
```

Figure 31: Task $d19$: Lift the cube 20cm above the table and 15 cm backward.

```
import torch
from torch import Tensor
from typing import Tuple

def generate_goal_pose(cube: Tensor) -> Tensor:
    """This function returns an eligible goal pose satisfying the following table top rearrangement task: "move the cube at 20cm to the
left of its initial position."

    The setup is 1 table of dimension 0.81 x 1 x 0.795 on which stands 1 cube of edge 5cm and having its origin in its center.
    The global origin (0,0,0) is located on the floor below the middle of one of the table's sides.
    X coordinates range: (0., +0.70) = (Bottom, Top)
    Y coordinates range: (-0.4, +0.4) = (Right, Left)
    The minimum Z value for objects located on the surface of the table is 0.795 + 0.05/2 = 0.82

    Args:
      - Tensor: a tensor of shape (3) representing the cube position on the table

    Returns:
      - Tensor: a tensor of shape (3) representing the goal position on the table
    """
    # Calculate the goal position by moving 20cm to the left of the initial position
    goal_position = torch.tensor([0.2, cube[1] - 0.2, 0.82])
    return goal_position
```

Figure 32: Task $d26$: Move the cube at 20cm to the left of its initial position.

```
import torch
from torch import Tensor
from typing import Tuple

def generate_relative_goal_poses(cube1: Tensor, cube2: Tensor, cube3: Tensor) -> Tuple[Tensor, Tensor, Tensor]:
    """This function returns a tuple of elligible goal poses satisfying the following table top rearrangement task: "rearrange three
cubes in such a way that the distance between each of them is 10 centimeters."

    The setup is 1 table of dimension 0.75 x 0.8 x 0.78 on which stands 3 cubes of edge 5cm.
    Each cube has its own origin in its center.
    The global environment origin (0,0,0) is located on the floor below the middle of one of the table's sides.
    X coordinates range: (0., +0.75) = (Bottom, Top)
    Y coordinates range: (-0.4, +0.4) = (Right, Left)
    The minimum Z value for objects located on the surface of the table is 0.78 + 0.05/2 = 0.805

    Args:
      - Tensor: a tensor of shape (3) representing the position of the first cube on the table
      - Tensor: a tensor of shape (3) representing the position of the second cube on the table
      - Tensor: a tensor of shape (3) representing the position of the third cube on the table

    Returns: Tuple(Tensor, Tensor, Tensor)
      - Tuple: a tuple of 3 tensors. Each tensor of shape (3) represents the goal position of one cube on the table
    """

    distance = 10 / 100  # Convert distance to meters
    if cube1[0] >= 0.5:
        cube1[0] = 0.5
    if cube1[1] >= 0.2:
        cube1[1] = 0.2
    if cube1[1] <= -0.2:
        cube1[1] = -0.2

    goal_positions = [cube1]

    for reference in [cube2, cube3]:
        found = False
        while not found:
            angle = torch.rand(1, device=reference.device) * 2 * torch.pi
            x_offset = distance * torch.cos(angle)
            y_offset = distance * torch.sin(angle)
            candidate_position = goal_positions[-1] + torch.tensor([x_offset, y_offset, 0], device=reference.device).squeeze()

            # Check if candidate_position is within table boundaries
            if (0 < candidate_position[0] < 0.75) and (-0.45 < candidate_position[1] < 0.45) and candidate_position[2] >= 0.8:
                all_distances_valid = True

                for pos in goal_positions:
                    dist = torch.norm(candidate_position - pos)
                    if not torch.isclose(dist, torch.tensor(distance, device=reference.device), rtol=1e-2, atol=1e-2):
                        all_distances_valid = False
                        break

                if all_distances_valid:
                    found = True
                    goal_positions.append(candidate_position)
            else:
                print(candidate_position)

    return tuple(goal_positions)
```

Figure 33: Task 29: Rearrange three cubes in such a way that the distance between each of them is 10 centimeters.

```python
import torch
from torch import Tensor
from typing import Tuple

def generate_goal_poses() -> Tuple[Tensor, Tensor, Tensor]:
    """
    This function returns an eligible goal pose satisfying the following table top rearrangement task: "Move the three cubes on the
table so at the end they form a right-angled triangle wih one corner at the center of the table."

    The setup is 1 table of dimension 0.75 x 0.8 x 0.78 on which stands 3 identical cubes of edge 5cm each and having their origins in
their respective centers.
    The global origin (0,0,0) is located on the floor below the middle of one of the table's sides.
    X coordinates range: (0., 0.75)
    Y coordinates range: (-0.4, 0.4)
    The minimum Z value for objects located on the surface of the table is 0.78 + 0.05/2 = 0.805

    Returns a tuple of Tensors:
     - [Tensor, Tensor, Tensor]: Each tensor is of shape (3) and contains a goal position
    """

    # Define the positions of the three cubes
    pos1 = torch.tensor([0.45, -0.25, 0.805])
    pos2 = torch.tensor([0.65, 0.25, 0.805])
    pos3 = torch.tensor([0.35, 0.25, 0.805])

    # Rotate the positions to form a right-angled triangle with one corner at the center of the table
    angle = torch.tensor([0, 0, -45]) * (3.14159 / 180)
    rot_mat = torch.tensor([[torch.cos(angle[2]), -torch.sin(angle[2]), 0],
                            [torch.sin(angle[2]), torch.cos(angle[2]), 0],
                            [0, 0, 1]])
    center_pos = torch.tensor([0.375, 0, 0.805])
    pos1 = torch.matmul(rot_mat, pos1 - center_pos) + center_pos
    pos2 = torch.matmul(rot_mat, pos2 - center_pos) + center_pos
    pos3 = torch.matmul(rot_mat, pos3 - center_pos) + center_pos

    return pos1, pos2, pos3
```

Figure 34: Task $d30$: Move the three cubes on the table so at the end they form a right-angled triangle with one corner at the center of the table.

A.4   EXAMPLES OF TASK DEPENDANT REWARD FUNCTIONS GENERATED BY LARG[2]

```python
import torch
from torch import Tensor
from typing import Tuple
from gpt.utils import *
from envs.utils.torch_jit_utils import *

def compute_franka_reward_generated( lfo_dist_reward: float, object_pos: Tensor,
    object_z_init: float, goal_dist_reward_scale: float, goal_bonus_reward_scale: float
) -> Tuple[Tensor, float]:
    """
    Compute the reward signal for a Franka Move task in Isaac Gym.
    Args:
        lfo_dist_reward: Distance reward between left finger and object.
        object_pos: Position of the object.
        object_z_init: Initial height of the object.
        object_dist_reward_scale: Scaling factor for the object distance reward.
        goal_dist_reward_scale: Scaling factor for the goal distance reward.
        goal_bonus_reward_scale: Scaling factor for the goal bonus reward.
    Returns:
        A tuple containing:
            - rewards: The reward signal for the current state.
            - og_d: The distance between the object pos and the goal pos
    """
    # Initialize goal position variable
    goal_pos = torch.tensor([0.7, 0.4, 0.82])

    # Object above table
    object_above = (object_pos[:, 2] - object_z_init) > 0.015

    # Object to goal reward
    og_d = torch.norm(goal_pos - object_pos, p=2, dim=-1)
    og_dist_reward = torch.zeros_like(lfo_dist_reward)
    og_dist_reward = torch.where(object_above, 1.0 / (0.04 + og_d), og_dist_reward)

    # Bonus if object is near the goal
    og_bonus_reward = torch.zeros_like(og_dist_reward)
    og_bonus_reward = torch.where(og_d <= 0.04, og_bonus_reward + 0.5, og_bonus_reward)

    # Total reward
    rewards = goal_dist_reward_scale * og_dist_reward  + goal_bonus_reward_scale * og_bonus_reward

    return rewards, og_d
```

Figure 35: Task $m02$: Move a cube to the top left corner of the table.

```python
import torch
from torch import Tensor
from typing import Tuple
from gpt.utils import *
from envs.utils.torch_jit_utils import *

def compute_franka_reward_generated( lfo_dist_reward: float, object_pos: Tensor,
    object_z_init: float, goal_dist_reward_scale: float, goal_bonus_reward_scale: float
) -> Tuple[Tensor, float]:
    """
    Compute the reward signal for a Franka Move task in Isaac Gym.
    Args:
        lfo_dist_reward: Distance reward between left finger and object.
        object_pos: Position of the object.
        object_z_init: Initial height of the object.
        object_dist_reward_scale: Scaling factor for the object distance reward.
        goal_dist_reward_scale: Scaling factor for the goal distance reward.
        goal_bonus_reward_scale: Scaling factor for the goal bonus reward.
    Returns:
        A tuple containing:
            - rewards: The reward signal for the current state.
            - og_d: The distance between the object pos and the goal pos
    """
    # Initialize goal position variable
    goal_pos = torch.tensor([0.7, 0.4, 0.82])

    # Object above table
    object_above = (object_pos[:, 2] - object_z_init) > 0.015

    # Object to goal reward
    og_d = torch.norm(goal_pos - object_pos, p=2, dim=-1)
    og_dist_reward = torch.zeros_like(lfo_dist_reward)
    og_dist_reward = torch.where(object_above, 1.0 / (0.04 + og_d), og_dist_reward)

    # Bonus if object is near the goal
    og_bonus_reward = torch.zeros_like(og_dist_reward)
    og_bonus_reward = torch.where(og_d <= 0.04, og_bonus_reward + 0.5, og_bonus_reward)

    # Total reward
    rewards = goal_dist_reward_scale * og_dist_reward  + goal_bonus_reward_scale * og_bonus_reward

    return rewards, og_d
```

Figure 36: Task $m04$: Move a cube at 20cm above the center of the table.

