# OpenReview forum: "LARG2, Language-based Automatic Reward and Goal Generation"
_ICLR.cc/2024/Conference — Submitted to ICLR 2024_

### Official Review · Reviewer_j1KR · 2023-10-18

**Soundness:** 2 fair
**Presentation:** 2 fair
**Contribution:** 2 fair
**Rating:** 3
**Confidence:** 4

**Summary:**

This paper proposes to leverage LLMs to generate goals and to generate code to implement reward functions in the context of a robotic environment. It looks at the impact of in-context examples and proposes automatic correction mechanisms to refine generated code.

**Strengths:**

I think the idea of using LLM to generate program-based reward functions and goals is good and will be useful for future developments in RL and autonomous learning in general.

**Weaknesses:**

The main weaknesses of this paper are along the following dimensions:

* a couple of very similar works are not discussed here.
* the evaluation is restricted to a single robotic setup with small diversity of goals and tasks considered
* the focus on examples seems superfluous as the results don't seem to indicate that it positively impacts performance
* the structure is hard to follow and there seem to be several ad-hoc parts that makes it not-trivial to apply this approach to other domains out of the box.

-----

Let's first discuss related works. There are other related approaches that are missing here and could further motivate the proposed approach.

1/ the paradigm of learning from human preference (Christiano’s paper at DeepMind on learning to do a backflip) attempts to circumvent the difficulty of hand-designing rewards, but is also limited by the need for much human feedback: this could be another argument to motivate the proposed approach?

2/ Other approaches propose to use LLM to generate goals!
  * https://arxiv.org/abs/2305.12487 uses an LLM to generate goals, hindsight relabels and rewards but without generating code
  * https://arxiv.org/abs/2302.06692 uses an LLM to generate exploration goals and computes rewards with the similarity of embeddings of the goal and a caption of the transition.
  * https://arxiv.org/pdf/2305.16291.pdf as well? or maybe they use the LLM to generate policies only.

Calling an LLM to generate rewards is slow and not very robust. Your approach proposes to improve on these aspects by generating reward functions as code, which is much faster to execute, and which you can refine.

3/ The origin of this line of work is the idea of reward machines: https://arxiv.org/abs/2010.03950. It seems that there are other papers proposing to use LLMs to generate reward functions in addition to Yu’s paper: https://arxiv.org/abs/2309.11489 and https://arxiv.org/abs/2303.00001, could you position the proposed approach with respect to these?

The references for goal-conditioned learning seem really specific. The usual  goal-conditioned references are Tom Schaul’s UVFA paper: http://proceedings.mlr.press/v37/schaul15.pdf (first goal-conditioned DRL algorithm), or Kaelbling 93 (pre-deep RL): https://citeseerx.ist.psu.edu/document?repid=rep1&type=pdf&doi=6df43f70f383007a946448122b75918e3a9d6682. There are also more recent reviews of goal-conditioned RL / autotelic RL: https://arxiv.org/abs/2012.09830. Huang 2022 is a weird reference to use on its own to introduce gcrl, while plappert, nair and openai’s papers also focus on very specific types of goals (target states in robotic tasks).

----

Now about the evaluation of the approach. The argument is that LLM can be used to automate the generation of large amounts of complex reward functions. The studied goals and tasks here are however quite limited in terms of difficulty and diversity: it's only about placing blocks in various 3D positions. These functions are very easy to code by hand and require only limited physical understanding. This approach should be evaluated on several setups, using varied goals (not only pushing blocks), including navigation goals, manipulation goals, crafting goals.

Having half of the reward function pre-coded is also weird, these bonuses might be counterproductive if we start to think of other goals the agent could do here (eg throwing blocks away, drawing an 8 with the gripper). It sounds like the approach remains very custom to the environment and I'm not sure how much is gained here in terms of engineering efforts vs reward functions outputs.

----

About the use of examples.
The paper says: “ table 1 which highlight a positive impact of supplemental examples using the GPT4 model” but it does not. GPT’s results are about the same with and without the examples, each beating the other in about half of the tasks. Plus it’s not clear how significant are these results or even what they report exactly? Is each number the percentages of goals generated that matched the description according to hard-coded validation functions? If so, how many goals does each point represent? Are these differences statistically significant? It sounds like we’re measuring two bernoulli probabilities, there are ways to test for significant differences in their parameters. What this table shows, how these numbers are computed, and their significance should be discussed. It sounds that there is no significant impact of having examples here, so this whole process of retrieval seems superfluous.

I feel like the example addition is more about in-context learning and less about chain of thought. CoT is originally about querying the LLM several times in a row and prompting it to generate reasoning before eventually performing the task (eg answering a question). There is a zero-shot version of that using the let’s think step by step. This is orthogonal to the question of including examples in the prompt. Here it does not seem the model is nudged to perform this chain-of-thought?

Table 2 similarly seems inconclusive. Every column sometimes performs best and it’s not clear that any of the columns performs significantly better than the no-example L column. The process of data collection and retrieval seems rather complicated and it's hard to get a sense of what's going on (eg where are examples of the examples retrieved?). Why doing all this if it brings no significant performance gains?

----
I think there is a confusion between tasks and goals here. In my understanding, a task refers to a specific MDP, so different tasks can vary in terms of any component of the MDP (eg state space, transition function). Goals are both a description of the desired outcome and a reward function measuring progress towards it (http://proceedings.mlr.press/v37/schaul15.pdf, https://arxiv.org/abs/2012.09830). So multi-goal problems are a subset of multi-task problems where the tasks vary in terms of the reward function + goal embeddings.

Under that framework, the proposed approach may become clearer. Both methods actually tackle multi-goal problems: one generates goal representations as target states and assumes corresponding distance-based reward functions, while the other keeps linguistic goal representations and generates the corresponding reward functions. The considered goals / tasks presented in the paper are indeed kind of the same. For instance putting the 'block on the right side' could be solved by either generating a target state + using a hard-coded distance-based reward functions or by keeping the linguistic goal representation and using a custom reward functions comparing the position of the block to the left-ride dividing line. In both case the agent is then conditioned on the goal representation (either linguistic directly or the generated target state) and gets rewarded by the reward function (either generated or assumed to be a function of the distance to the target state). I think presenting it this way makes it much clearer and may avoid repetitions.

Presenting it this way + removing the example part that does not seem to bring significant advantage would make the whole presentation much clearer and would also simplify Fig1.

----
I found the paper overall not very clearly written. The main paper itself gives little sense of what's going on, how reward functions look like or how the correction mechanism works, these very important things are left to the appendix.


----
Minor comments:
* the opening quotes are not well-formated, consider using `` instead of “.
* “Generate n paraphrases for the task bellow”--> ‘below’
* I_b is not defined in table 2
* the prompt examples in the appendix have very low resolution


---- Conclusion

I think the idea is good, but it could be better framed and presented. The final results are not very diverse so it does not convince me that anyone should really use this approach for reward function generations. I'd also like to see a discussion of other papers proposing LLM-based generation of reward functions.

**Questions:**

Is there no correction mechanism when tests are not passed? Is improvement only applied in case of an execution error? (Figure 4).

---

> ### Author Response · Authors · 2023-11-21
> **acknowledgment**
>
> Dear Reviewer,
>
> We would like to thank you for taking the time to review our work and provide valuable comments and suggestions.
>
> We acknowledge the importance of your input and would like to assure you that all your comments will be carefully considered to prepare a more mature version of our paper.
>
> We will fix the issues that you spotted and revise the description of our method to make it clearer. Adding too many elements of the whole process seems to have created confusion and weakened the discourse.
> We will split this content and focus on individual contributions which should allow space for better method descriptions and more extensive experimental support.
>
> Your expertise and critical analysis have played an essential role in strengthening the quality of our communication.
>
> We are committed to addressing all the raised points and look forward to resubmitting an improved version of our work in the near future.
>
> With gratitude and best regards,
>
> The authors

---

### Official Review · Reviewer_zRqc · 2023-10-29

**Soundness:** 2 fair
**Presentation:** 1 poor
**Contribution:** 2 fair
**Rating:** 3
**Confidence:** 4

**Summary:**

The paper proposes an LLM-based approach to convert text based task descriptions into reward and goal generating functions. The effectiveness of the proposed method is evaluated via successful training of policies in robotic manipulation tasks.

**Strengths:**

The proposed method seems to be novel and is relevant.

**Weaknesses:**

Perhaps more details, including the scope, limitations, high level idea etc., of the paper needs to be provided in the introduction.   The methodology repeatedly uses duplicate notations which hinders the ease of understanding of the paper.

**Questions:**

1.	What are the limitations on the complexity of the goals? Could the approach be used in tasks with non-goal based reward functions (eg: training an agent to move in a circular pattern/do a backflip etc.,)?

2.	Several notations in 4.1 are duplicated. Eg: R, S, etc., have previously been used for RL related terms. Similarly, in 4.2.1, p was previously used in the transition function in Section 2,  and T was the notation used for the transition function.

3.	Regarding the code-validation described in Section 4.3, does it only ensure that the code runs without issues or does it also ensure that the reward function is valid/close to optimal?

4.	With reference to Table 2, how are the reward functions ranked?

5.	The limitations section don’t really mention the limitations clearly. The way it is written, it gives the impression that all limitations are addressed by the auto-correction loop.

6.	Notation G is duplicated (Guidelines and Goal space)

7.	In section 4, first paragraph, it is better to mention the kind of input gathered for the second step

8.	Is there a way to ensure that the reward function is indeed close to ideal?

9.	Several typos/grammatical errors – ‘rewards functions’ in conclusion. ‘if the code fail..’, ‘can not be guarantee’, ‘commonly encounter..’ in Section 4.3, Section 4.1 – ‘bellow’ should be ‘below’

10. I believe Fig 15 on page 5 is supposed to be Fig 2.

---

> ### Author Response · Authors · 2023-11-21
> **acknowledgment**
>
> Dear Reviewer,
>
> We would like to thank you for taking the time to review our work and provide valuable comments and suggestions.
>
> We acknowledge the importance of your input and would like to assure you that all your comments will be carefully considered to prepare a more mature version of our paper.
>
> We will fix the issues that you spotted and revise the description of our method to make it clearer. Adding too many elements of the whole process seems to have created confusion and weakened the discourse.
> We will split this content and focus on individual contributions which should allow space for better method descriptions and more extensive experimental support.
>
> Your expertise and critical analysis have played an essential role in strengthening the quality of our communication.
>
> We are committed to addressing all the raised points and look forward to resubmitting an improved version of our work in the near future.
>
> With gratitude and best regards,
>
> The authors

---

### Official Review · Reviewer_Ev2L · 2023-10-29

**Soundness:** 2 fair
**Presentation:** 1 poor
**Contribution:** 2 fair
**Rating:** 3
**Confidence:** 4

**Summary:**

This paper introduces LARG2, a method for converting text-based task descriptions into corresponding reward and goal-generation functions. LARG2 leverages context, environment, guidelines, and task descriptions to query a code database to retrieve function examples. Then, these examples are used to convert textual task descriptions into goal poses or reward functions.

**Strengths:**

This paper studies the pertinent problem of reward design and goal-generation in multi-goal/multi-task reinforcement learning. It investigates an interesting approach of combining LLM with database search as a solution to the problem. I believe that this approach is new in the growing space of LLM-aided problem design for decision making and could be impactful for the community.

**Weaknesses:**

1. This paper is very difficult to read. The text and code snippet in figures are too small to read. It would be helpful to include one illustrative example in the main text. The method section also interleaves method explanation with implementation particular to the pick-and-place tasks in the experiment section; this makes it difficult to understand how general LARG2 actually is.

2. LARG2 requires a lot of assumptions to work. The goal and reward functions are not generated from the LLM in the traditional sense. Instead, LLM simply samples Cartesian coordinates in the goal generation case and composes a reward function from existing reward components provided in the simulator. It's unclear how these methods would transfer to new tasks that do not come with existing assets that allow LARG2 to perform it search and retrieval procedure.

3. The experiments seem preliminary. They are carried out only in a single pick-and-place setup. There are no baseline comparison. The paper also lacks any ablation study besides choices of LLMs; given the fairly involved and complicated algorithmic design, an ablation study would be very helpful to better understand the importance of various LARG2 components.

**Questions:**

1. An external baseline should be included. Comparison to a human-written reward function would help ground the performance of LARG2.

2. Ablation studies should be included.

3. More robot tasks and morphologies should be considered.

4. Could the goals in Figure 7 be overlayyed on top of the actual table top environment? This will make interpreting the figure much more easier.

---

> ### Author Response · Authors · 2023-11-21
> **acknowledgment**
>
> Dear Reviewer,
>
> We would like to thank you for taking the time to review our work and provide valuable comments and suggestions.
>
> We acknowledge the importance of your input and would like to assure you that all your comments will be carefully considered to prepare a more mature version of our paper.
>
> We will fix the issues that you spotted and revise the description of our method to make it clearer. Adding too many elements of the whole process seems to have created confusion and weakened the discourse.
> We will split this content and focus on individual contributions which should allow space for better method descriptions and more extensive experimental support.
>
> Your expertise and critical analysis have played an essential role in strengthening the quality of our communication.
>
> We are committed to addressing all the raised points and look forward to resubmitting an improved version of our work in the near future.
>
> With gratitude and best regards,
>
> The authors

---

### Official Review · Reviewer_S7zE · 2023-10-31

**Soundness:** 1 poor
**Presentation:** 1 poor
**Contribution:** 2 fair
**Rating:** 3
**Confidence:** 3

**Summary:**

The authors propose "Language-based Automatic Reward and Goal Generation" ($LARG^2$) to do automatic reward generation with LLM for goal-conditioned and multi-task RL problems. $LARG^2$ incorporates an automatic code correction loop and a supplemental code example database for CoT inferences to improve the performance. The proposed method achieves good success rate in a simulated tabletop robot arm environment.

**Strengths:**

- The proposed method is able to complete most of the proposed tasks with a decent success rate (assuming the performance in Table 1 and 2 is the success rate).

**Weaknesses:**

- In general, the writing and formatting of the paper are subpar.
The method description (Sec 4) is not well-structured or motivated. It feels more like a step-by-step explanation of what happened and a lot of the details can be left out for the sake of clarity. Personally, I also find using single characters to denote a large variety of things (prompts, text descriptions, codes, functions, sets, etc) confusing and hard to follow.
- There are no visuals of the experiment setup nor any detailed descriptions of the scene. From the description in the paper, it's very hard for me to evaluate the difficulties of the tasks.
- The contribution of this work is very questionable. From my understanding, these are the claimed contributions:
    - An automatic process of code correction with LLMs and feedback from the code interpreter. This is not a new concept at all and can be considered more or less a standard procedure in LLM-assisted code generation, for a reference see [1]. From the description in this paper, I don't see any major differences from existing methods.
    - Using supplemental code examples collected from open-source repositories for CoT. Furthermore, the baseline method $L$ without supplemental examples in Table 2 still performs reasonably well compared to $LARG^2$.
    - Strong empirical performances in GC and MT tasks. However, there are no baseline methods to compare with in the experiment section. It is therefore very hard to evaluate the proposed method.

Minor mistakes and typos:

- In the abstract, the abbreviation "LLM" is mentioned before the full name.
- Second to the last line of Sec 3.4, "Chain-of-though" should be "Chain-of-Thought".
- Texts in Figure 3 are too small to read. Consider using a larger font size.
- First line of Sec 4.2.1, "re-arranging a set of objects composing the scene". Do you mean "re-arranging a set of objects in the scene"?

**Questions:**

- How are the supplemental example codes utilized in the prompting? It's surprising that this part is not described at all in the main text despite being one of the major contributions.
- How is the "performance" in Table 1 and 2 measured?
- What is the purpose of Figure 7? What is the information I am suppose to get from the generated goal positions?

---

> ### Author Response · Authors · 2023-11-21
> **acknowledgment**
>
> Dear Reviewer,
>
> We would like to thank you for taking the time to review our work and provide valuable comments and suggestions.
>
> We acknowledge the importance of your input and would like to assure you that all your comments will be carefully considered to prepare a more mature version of our paper.
>
> We will fix the issues that you spotted and revise the description of our method to make it clearer. Adding too many elements of the whole process seems to have created confusion and weakened the discourse.
> We will split this content and focus on individual contributions which should allow space for better method descriptions and more extensive experimental support.
>
> Your expertise and critical analysis have played an essential role in strengthening the quality of our communication.
>
> We are committed to addressing all the raised points and look forward to resubmitting an improved version of our work in the near future.
>
> With gratitude and best regards,
>
> The authors

---

### Meta-Review · Area_Chair_okjj · 2023-12-05

**Metareview:**

The paper proposes Language-based Automatic Reward and Goal Generation (LARG$^2$), a framework that uses large language models (LLMs) to generate functional specifications of goals given a text-based representation of a task. The goals correspond to reward functions for reinforcement learning. The framework includes mechanisms to automatically detect and correct errors in the generated code, and a method that performs retrieval from a database of code examples to improve performance.

The paper received four reviews that largely agree on the paper's strengths and weaknesses. At least three reviewers find that the paper addresses an important problem, namely the automatic generation of goals (and rewards) for reinforcement learning. However, the reviewers agree that there are a number of fundamental issues with the paper in its current form. Among them, reviewers question the significance of the paper's contributions both in terms of the framework and the results. The underlying methods themselves are not new and the experimental results do not provide empirical evidence that the framework provides much in terms of gains. This is partially due to the preliminary nature of the experiments that consider simple domains with little diversity, that do not include comparisons to appropriate baselines (e.g., the performance on GC and MT tasks is difficult to judge without baseline comparisons), and that do not ablate the different components of the framework. Further, there are questions about the statistical significance of some of the results, which seem inconclusive despite claims otherwise.

**Justification For Why Not Higher Score:**

There is consensus among the reviewers that the paper should be rejected.

**Justification For Why Not Lower Score:**

N/A

---

### Decision · Program_Chairs · 2024-01-16

Reject